



# Long-Term Impact of Cover Crop and Reduced Disturbance Tillage on Soil Pore Size and Soil Water Storage

Samuel N. Araya[1], Jeffrey P. Mitchell[2], Jan W. Hopmans[3], and Teamrat A. Ghezzehei[4]

[1]Earth System Science, Stanford University, Stanford, CA, USA
[2]Department of Plant Sciences, University of California, Davis, CA, USA
[3]Department of Land, Air and Water Resources, University of California, Davis, CA, USA
[4]Life and Environmental Science, University of California, Merced, CA, USA

*Correspondence to*: Samuel N. Araya (araya@stanford.edu)

**Abstract.** Using laboratory measurements and numerical simulations, we studied the long-term impact of contrasting tillage
and cover cropping systems on soil structure and soil hydraulic properties. Complete water retention and conductivity curves
for top (0 – 5 cm) and subsurface (20 – 25 cm) samples were characterized and contrasted. Plot-level properties of water
storage and retention were evaluated using numerical simulations in HYDRUS-2D software. Soils under no-till (NT) and cover
cropping (CC) systems showed an improved soil structure in terms of pore size distribution (PSD) and the hydraulic
conductivity (K) under these systems led to increased infiltration rate and water retention. The conventional measurement of
water content at field capacity (water content at -33 kPa suction) and the associated plant available water (PAW) showed that
NT and CC plots had lower water content at field capacity and lower PAW compared to standard-till (ST) and plots without
cover crop (NO). The numerical simulations, however, showed that NT and CC plots have higher profile-level water storage
(albeit marginal in magnitude) and water availability following irrigation. Because the numerical simulations consider
retention and conductivity functions simultaneously and dynamically through time, they allow the capture of hydraulic
properties that are arguably more relevant to crops. The changes in PSD, water conductivity, and water storage associated with
NT and CC systems observed in this study suggest that these systems are beneficial to general soil health and improve water
retention at the plot scale.

**List of Acronyms and Symbols**

CC, Cover crop; HCF, Hydraulic conductivity function; NT, No-till; PAW, Plant available water content; PSD, Pore size
distribution; ST, Standard-till; WRC, Water retention curve



$K_s$, Saturated hydraulic conductivity; $\theta_{FC}$, Volumetric water content at field capacity; $\theta_{PWP}$, Volumetric water content at permanent wilting point (-15 MPa suction); $\rho_b$, Bulk density; $h$, negative water suction ($h = -\psi$); $K$, Hydraulic conductivity; $\theta$, Volumetric water content; $\psi$, Matric potential.

## 1 Introduction

Improving soil health—the vitality of a soil in sustaining the socio-ecological functions of its enfolding land (Janzen et al., 2021)—is one of the main challenges of our time as we grapple with the demands of growing population and changing climate. The tools at our disposal to achieve this goal in agricultural lands are collectively known as conservation agriculture practices. Conservation agriculture is characterized by a combination of three linked principles: (1) reduced mechanical soil disturbance, (2) preservation of a permanent organic soil cover, and (3) diversification of crop species (Kassam et al., 2019; Li et al., 2018;

Mitchell et al., 2019). The adoption of conservation agriculture has been growing worldwide at an increasing rate since the 1960s. Between 2008 and 2015, the global area under conservation agriculture grew by 69% to 180 M ha (Blanco-Canqui and Ruis, 2018; Kassam et al., 2019). In California's highly productive Central Valley region the cultivated area under conservation agriculture for tomato and corn production has increased from less than 5,000 ha in 2004 to over 140,000 ha in 2012 (Mitchell et al. (2016a).

Conservation agriculture promises two main categories of benefits to soil health and soil functions. First, conservation agriculture, specifically, reduced tillage, eliminates the negative effects associated with standard (conventional) tillage (ST), including degradation of soil structure, erosion, loss of nutrients, and reduction in soil microbial diversity and soil organic matter (Lal et al., 2007; Zuber and Villamil, 2016). Second, conservation agriculture supports the development of healthy soils. For example, reduced disturbance tillage systems have been shown to sequester carbon and decrease greenhouse gas emission

(Reicosky and Allmaras, 2003; Palm et al., 2014; Sanz-Cobena et al., 2017); increase soil fertility (Veenstra et al., 2006, 2007); increase microbial biomass, richness, and activity (Zuber and Villamil, 2016; Martens, 2004; Johnson and Hoyt, 1999); and improve environmental quality (Baker et al., 2005; Madden et al., 2008; Reicosky and Allmaras, 2003) without compromising yield (Naab et al., 2017; Rasmussen, 1999; Alvarez and Steinbach, 2009) while reducing cost (Upadhyaya et al., 2001; Mitchell et al., 2009; González-Sánchez et al., 2016). Cover cropping—planting between cropping seasons to maintain soil coverage

throughout the year and often to replenish soil N—provides many beneficial services including soil cover, residues, and biological diversity (Mitchell et al., 2019). Cover crops have been shown to reduce erosion (Reicosky and Forcella, 1998; Shelton et al., 2000) diseases, and pest pressure (Mitchell et al., 2017); while increasing soil fertility (Büchi et al., 2018; Abdalla et al., 2019), as well as microbial biomass, richness, and activity (Fernandez et al., 2016; Duchene et al., 2017).

Conservation agriculture is also credited with myriad beneficial changes to soil hydrology, including increases in
macroporosity (Abdollahi et al., 2014; Burr-Hersey et al., 2017), water storage (Liu et al., 2019; Basche et al., 2016a; Duchene et al., 2017; Finney et al., 2017; Ashworth et al., 2017), and infiltration (Hudson, 1994; Johnson and Hoyt, 1999; Basche and DeLonge, 2017). Mitchell (2017) found cover crops increased infiltration by 2.8 times compared to soils without cover crops. Based on a meta-analysis from 27 studies, Basche and DeLonge (2017) concluded that cover cropping was effective in enhancing soil water storage and other soil hydrologic properties when practiced for longer-term (> 10 years) and in drier environments (< 9000 mm annual rainfall).

However, conservation agriculture can also lead to undesired negative outcomes. For example, reduced tillage systems can cause soil consolidation and compaction that can reverse the beneficial physical soil health outcomes (Blanco-Canqui and Ruis, 2018). Several studies have noted the critical lack of field studies and the need for evaluation of long-term effects of conservation agriculture on the soil physical and hydraulic properties and soil hydrological processes (Peña-Sancho et al., 2016; Basche and DeLonge, 2017; Blanco-Canqui and Ruis, 2018; Bacq-Labreuil et al., 2019). In this study, we planned to assess the long-term individual and interactive impacts of reduced tillage and cover crops practices on soil structure and associated hydrologic soil functions. We evaluated the properties of soil cores collected from the California Conservation Agricultural Systems Innovation (CASI) Center, where plots have been under a mix of reduced tillage and cover crop treatments since 1999. Specifically, we aimed to test whether conservation agriculture results in significant alterations in water retention, pore size distribution, density, hydraulic conductivity as well as static and dynamic field capacity.

## 2 Methods

### 2.1 Study site and experimental design

The CASI study site is located at the University of California West Side Research and Extension Center in Five Points, California (Figure 1). The experimental field has two-factor replicated treatments of tillage and winter cover cropping: standard-till with and without cover crops (ST-NO and ST-CC, respectively); and conservation (reduced disturbance) tillage with and without cover crops (NT-NO and NT-CC, respectively). CASI defines conservation tillage as a range of production practices that reduce primary intercrop tillage operations and either preserve 30% or more residue cover or reduce the total number of tillage passes by 40% or more (Mitchell, 2016). Throughout this manuscript, we will use the more descriptive no-till (NT) instead of conservation tillage.

Each treatment combination was replicated eight times in a randomized complete block implemented on a 9 by 82-m dimension plot with an approximately 10-m buffer guard between the tillage treatments. All tractor and implement traffic were restricted



to the furrows and planting beds were never moved. While the operations used varied from year to year, the number of tractor passes for the NT plots was always reduced by 40% or more relative to the ST plots (Mitchell et al., 2012).

85   The soil type at the study site is a Panoche clay loam (fine-loamy, mixed, superactive, thermic Typic Haplocambids) which is representative of much of California's Central Valley. For the first 12 years of the conservation agriculture experiment (between 2000 and 2012), tomato and cotton were grown in rotation, followed by a rotation of sorghum with garbanzo beans since 2012. All plots were irrigated by subsurface drip.

The cover crops were a mix of triticale (*Triticosecale Wittm*), cereal rye (*Secale cereale L.*), common vetch (*Vicia sativa*), radish (*Raphanus sativus*), and clover (*Trifolium incarnatum*) seeded in 20 cm rows at 89.2 kg ha- in late October. The cover
90   crops are terminated in late March of the following year using a stalk chopper followed by disk incorporation in the ST system or sprayed with a 2% solution application of glyphosate after chopping and left on the surface as a mulch in the NT systems. Detailed descriptions of the study site and management have been published in previous works (Mitchell et al., 2017, 2015; Veenstra et al., 2006; Mitchell et al., 2016b)

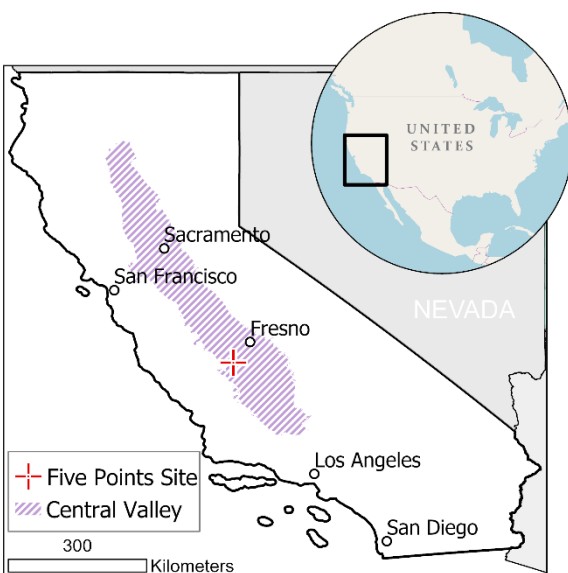

95   **Figure 1 Study site location at Five Points, California (California's Central Valley extent map from Faunt (2012)).**



## 2.2 Sampling

Sampling was done in mid-November 2017, months after tillage in the ST treatment plots to avoid the immediate effects of tillage since we were primarily interested in the long-term effects of the treatments. Tillage operations have a transitory effect on porosity and associated soil hydraulic properties as structures collapse, mainly driven by wetting and drying cycles post tillage (Or et al., 2000; Mapa et al., 1986). The immediate alterations of tillage on soil porosity and hydraulic properties have been shown to diminish rapidly following only a few wetting and drying cycles (Strudley et al., 2008; Alletto et al., 2015; Green et al., 2003).

Undisturbed soil samples from the top (0 – 5 cm) and subsurface (20 – 25 cm) layers were collected carefully using a 250 cm$^3$ volume sampling ring (8 cm diameter by 5 cm height). The depths were chosen to correspond with the depth disturbed by disking to incorporate residue in the ST plots (i.e., 0 – 20 cm depth) (Mitchell et al., 2015; Veenstra et al., 2006) and the deeper layer. Samples were collected along the strip ridges within the plots away from the trafficked furrows but slightly off-center to avoid drip irrigation tubes that were buried at the center of ridges. A total of 32 samples were collected by taking one surface, and one subsurface sample from four of the eight treatment replicate plots. This resulted in four replicates of surface and subsurface samples per treatment. The samples were stored at 4 °C before laboratory analysis.

## 2.3 Laboratory measurements

To measure the long-term impact of NT and CC practices on soil structure, we measured the soil bulk density ($\rho_b$), total porosity, pore size distribution (PSD), and soil hydraulic properties (water retention curve, WRC, and hydraulic conductivity function, HCF) of the soil cores.

The saturated hydraulic conductivity ($K_s$) was measured using the falling-head method. For this method, soils were saturated by immersing sample cores in degassed, 0.01 M CaCl$_2$ solution so that the water level was close to the rim. $K_s$ of the saturated soil was then measured by the falling-head method using the KSAT instrument (METER Group, Inc., Munich, Germany) by allowing a 5 cm column of degassed, 0.01 M CaCl$_2$ solution to flow through the soil core. The setup was such so that the flow direction was downward. Following the $K_s$ measurement, soil WRC, and HCF data were determined simultaneously using the evaporation method as developed for the HYPROP instrument (METER Group, Inc., Munich, Germany). The HYPROP simultaneously measures, at high frequency (10 min), suction inside the soil cores at two different depths along with weight loss while saturated soil cores dry. This allows for the calculation of WRC, $\theta(\psi)$, and HCF, $K(\psi)$. Following the HYPROP measurements, soil water retention in the range from $10^3$ to $10^6$ cm was determined by using the WP4C instrument (Decagon Devices, Inc, Pullman, WA, USA).



We use the conventional definition for field capacity ($\theta_{FC}$) and permanent wilting point ($\theta_{PWP}$) as the volumetric water content

with the corresponding volume of water retained in the soil at −33 kPa and −1,500 kPa suction, respectively. $\theta_{FC}$ and $\theta_{PWP}$ are approximations of water retained after internal drainage has ceased, and the soil water content limit at which plants cannot recover from turgidity, respectively (Hillel, 1998). We calculated plant available water (PAW) as the difference between $\theta_{FC}$ and $\theta_{PWP}$ (i.e., $PAW = \theta_{FC} - \theta_{PWP}$). In addition to the saturated hydraulic conductivity, we also calculated the unsaturated hydraulic conductivity near field capacity water content at -10 kPa.

Throughout this manuscript, the term water suction, $h$, is used to represent the soil water matric potential, $\psi$, such that $h = -\psi$ (cm).

### 2.4 Soil porosity determination

Total soil porosity ($P$) was calculated as $P = 1 - \rho_b/\rho_p$, where $\rho_p$ is the particle density of soil, taken as 2650 kg m$^{-3}$, and $\rho_b$ is the soil bulk density determined using the standard core method (Grossman and Reinsch, 2002).

The effective pore size distribution (PSD) was estimated from the slope of the WRC using the differential water capacity (Klute, 1986). For this, the WRC—$\theta(h)$—was first transformed into a curve of effective saturation ($S$) as a function of effective pore radius ($r$), $S(r)$. $S$ was calculated as $S = (\theta - \theta_r)/(\theta_s - \theta_r)$, where $\theta_s$ and $\theta_r$ are the saturated and residual volumetric water contents estimated from a bimodal constrained van Genuchten model fit (Durner, 1994) of measured WRC. The draining pore radius was approximated using the Young-Laplace equation (1):

$$r = \frac{2\gamma \cos(\beta)}{\rho_w g h} = \frac{1490}{h} \tag{1}$$

where $r$ [μm] is pore radius, $h$ [cm] is the suction, $\gamma$ is the surface tension between water and air (0.0729 N m$^{-1}$), $\beta$ is the contact angle (assumed 0), $\rho_w$ is the density of water (1000 kg m$^{-3}$), and $g$ is the acceleration due to gravity (9.81 m s$^{-2}$). The PSD curves were then calculated as (2):

$$f_p(\ln r) = -\frac{dS}{d \ln r} \tag{2}$$

where $f_p$ [-] is the density function of effective pore sizes. Prior to calculating PSD, the $S(r)$ curve was fitted with a cubic

smoothing spline to remove noise in the measurement data (Kastanek and Nielsen, 2001; Pires et al., 2008). For a deeper insight, we divided pore sizes into four ranges: intra-microaggregates (<0.2 μm), intra-aggregates (0.2 – 10 μm), small



macropores (10 – 50 µm), and large macropores (50 – 1000 µm). These range categories allowed us to perform statistical comparisons on the relative abundance of the pore size ranges among the different treatments.

**2.5 Soil water storage simulations**

To measure the interactive impact of changes in WRC and HCF on profile water dynamics and storage, we conducted a numerical simulation of field irrigation. The fate of irrigation water applied on the different treatment plots was simulated in HYDRUS-2D software where water flow is modeled using a modified form of the Richards' equation (Equation 3) which incorporates a sink term to account for water uptake by plant roots (Simunek et al., 2012).

$$\frac{\partial \theta}{\partial t} = \frac{\partial}{\partial x_i}\left[K\left(K_{ij}^A \frac{\partial h}{\partial x_j}\right)\right] - S_r \qquad (3)$$

where $\theta$ [L³L⁻³] is the volumetric water content, $t$ [T] is time, $x_i$ [L] are the spatial coordinates, $K$ [LT⁻¹] is the unsaturated hydraulic conductivity, $K_{ij}^A$ [-] are the components of a dimensionless anisotropy tensor, $h$ [L] is the pressure head, and $S_r$ [T⁻¹] is the sink term representing the rate of water volume removed due to plant water uptake.

The domain was set up as an axisymmetric cylinder of 18 cm radius and 100 cm depth. Figure 2 illustrates the model domain

sketch and the domain setup in HYDRUS-2D. The domain was discretized with 1473 nodes and 2788 triangular elements. This discretization mesh was refined to have more nodes around the emitter (0.5 cm spacing) and soil layer boundaries (1 cm spacing) to capture expected high rates of changes in soil moisture. The material distribution in terms of soil hydraulic properties was such that the top 0 – 20 cm and the subsurface 20 – 30 cm were that of those measured in this study (Section 3.3). Soil hydraulic properties for the bottom layers, 30 – 60 and 60 -100 cm layers, were predicted from soil characteristics

using Rosetta-H5 pedotransfer function (Schaap et al., 2001) and van Genuchten-Mualem (1980) hydraulic model. Soil characteristics for these layers were based on soil properties of C1 and C2 soil horizons (41 – 58 and 58 – 91 cm depths, respectively) for Panoche soils, Pedon ID S1978CA029001 (National Cooperative Soil Survey, n.d.).

Subsurface irrigation emitter was represented with a sphere of 1 cm radius buried 10 cm below the surface. We simulated the fate of 4.8 cm depth equivalent irrigation applied at an emitter discharge rate of 0.61 l h⁻¹ (0.60 cm h⁻¹ equivalent irrigation

depth) in each of the 16 sampled plots.

The entire domain surface area (1017.9 cm²) was associated with transpiration and the root water uptake was modeled by the HYDRUS-2D default Feddes' parameters for a tomato plant. Root spatial distribution was implemented using Vrugt et al.



(2001) functions with a maximum rooting depth of 35 cm, a maximum rooting radius of 15 cm, depth of maximum uptake intensity at 10 cm, and radius of maximum uptake intensity at 0 cm.

Atmospheric boundary condition was set for the surface layer and a free drainage lower boundary for the bottom layer. The atmospheric boundary condition was defined by transpiration which was calculated as the potential crop evapotranspiration based on Equation 4 (Allen et al., 1998).

$$ET_c = K_c \times ET_0 \qquad (4)$$

Where $ET_c$ [LT$^{-1}$]is potential crop evapotranspiration, $K_c$ [-] is crop coefficient (= 1.15 for tomato mid-season (Allen et al.,
1998), and $ET_0$ [LT$^{-1}$] is the reference potential evapotranspiration.

For the computation of crop evapotranspiration, daily reference potential evapotranspiration ($ET_0$) for a week during May 2018 (May 6 to 12) were acquired from the nearest weather station (CIMIS Five Points Station, https://cimis.water.ca.gov/).

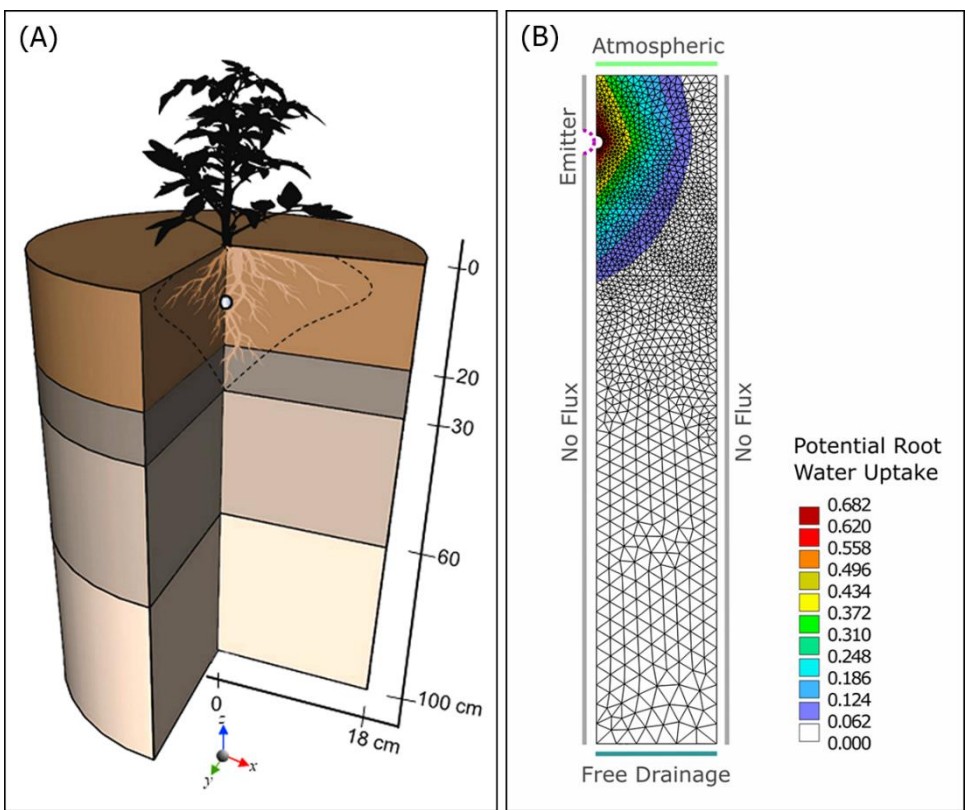

**Figure 2 (A) 3D schematic representation of the domain geometry and material distribution. (B) Domain setup in Hydrus-2D showing the finite element mesh, related boundary conditions, and potential root water uptake rate distribution.**

The starting pressure head of the entire model domain was set to -1000 cm and simulation was initialized by a 14-week spin-up period. The model was run recursively with 2.5 cm equivalent depth irrigation applied at the start of every week for 14 weeks after which the final simulation is run with 4.8 cm equivalent depth irrigation (at the rate of 0.6 cm h$^{-1}$ for 8 h) application (Figure 3). The amount of water retained in a given soil profile layer following irrigation is calculated as equivalent water depth changes using Equation 5.

$$\Delta W_t = W_t - W_{t0} \tag{5}$$

where $\Delta W_t$ [L] is equivalent water depth retained in the soil profile $t$ hours after irrigation application, $W_t$ is the equivalent water depth in the soil profile $t$ hours after irrigation, and $Wt_0$ is equivalent water depth immediately before irrigation application.





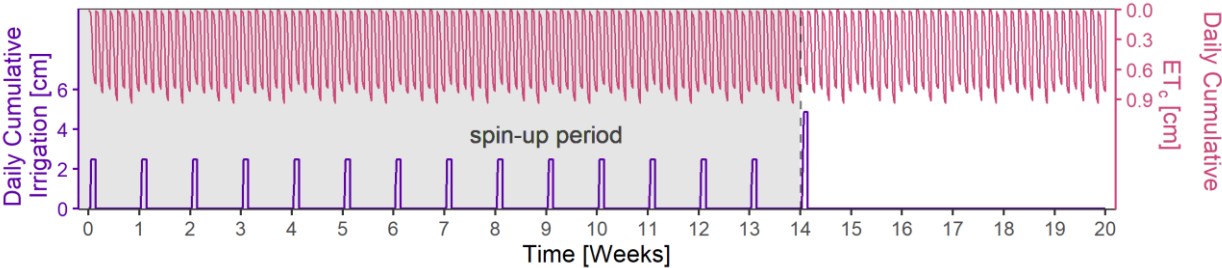

**Figure 3 Daily cumulative irrigation and potential crop evapotranspiration through the 14 weeks of spin-up period (grey background) and the final simulation.**

### 2.6 Statistical analysis

All quantitative results are expressed as means of four replicates ± standard error unless otherwise indicated. Differences in means were tested by analysis of variance (ANOVA) and pairwise comparison of treatments done using Tukey's honest significant difference (HSD) test at $p < 0.15$ significance level unless otherwise stated (Least Significant Difference table are provided in Appendix A Table B1). Hydraulic conductivity values were log-transformed before statistical analysis to make their distribution more normal. The normality of the data and the homogeneity of variances were checked using Shapiro–

Wilk's and Levene's tests, respectively. All statistical analyses were performed using R statistical software (R Core Team, 2019).

### 3 Results and Discussion

Water retention and conductivity properties were measured for each soil sample using the KSAT, HYPROP, and WP4C instruments. The saturated hydraulic conductivities were measured using KSAT, the unsaturated hydraulic conductivities and

the WRC using HYPROP and the water retention at extreme dry range using WP4C instruments (Figure 4). The HCF and WRC for all the samples are provided in Appendix A Figures A1 and A2.





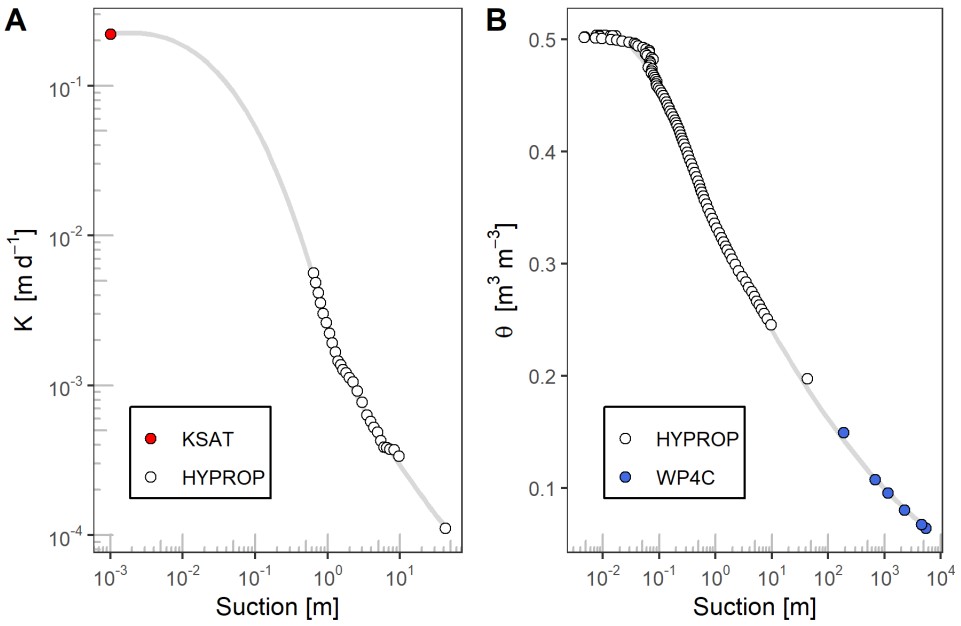

**Figure 4 Plot of measured hydraulic conductivity (A) and water retention (B) for one of the topsoil ST-CC samples with the measurement instrument labeled. Grey lines are LOESS smooth trend lines.**

## 3.1 Pore size distribution

The mean soil PSDs for the different systems are shown in Figure 5 (A). PSD curves for the individual samples are provided in Figure A3. A wider spread of PSD implies a heterogeneous mix of pore sizes and is indicative of soil with a more developed structure. The maximum pore volume density for the top soils occurred between sizes 15 and 20 µm diameter pores with the exception for NT-CC soils which showed a bimodal distribution with maximum pore volume density around 4 and 518 µm (Table 1).



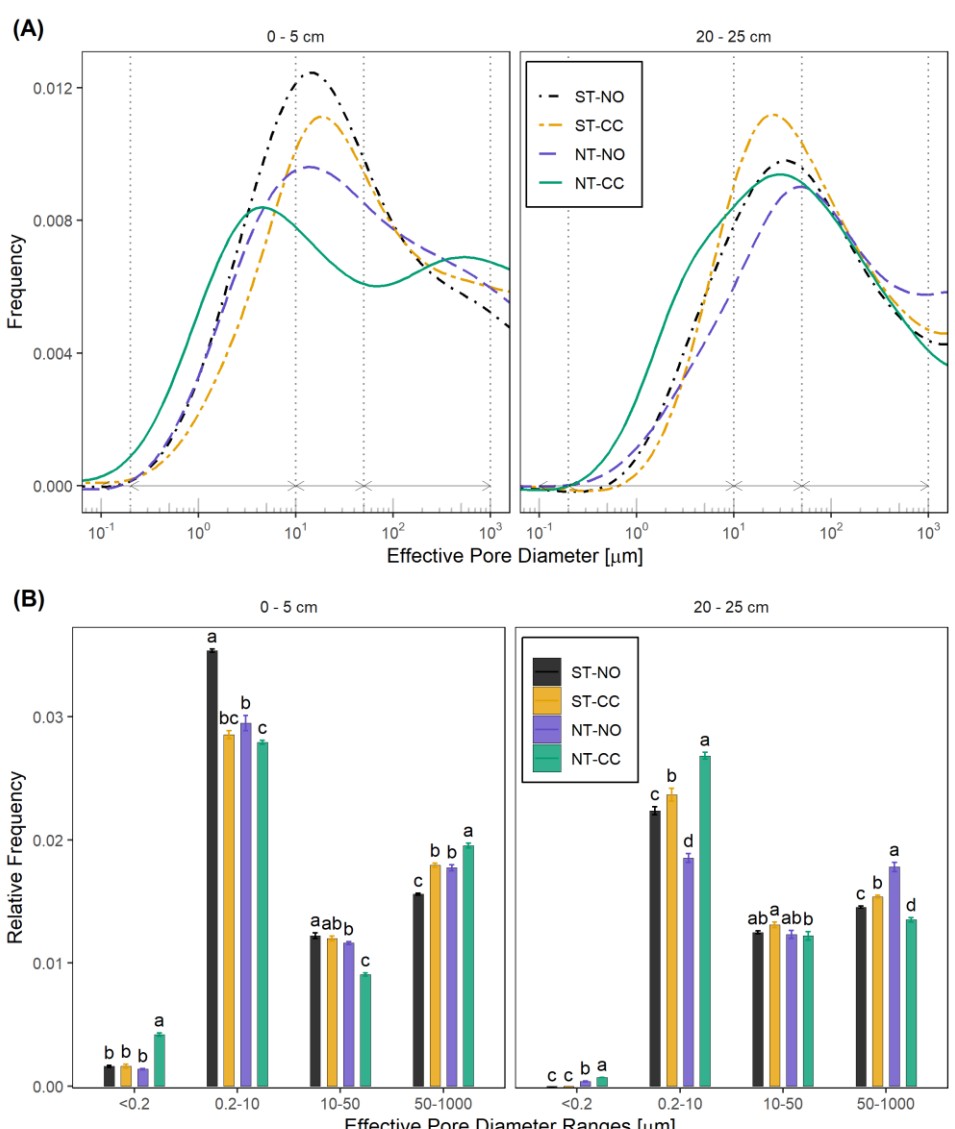

**Figure 5 (A) Pore size distribution for the top (0 – 5 cm) and subsurface (20 – 25 cm) soil layers. Dotted vertical lines and horizontal arrows indicate the characteristic pore diameter ranges of <0.2, 0.2 – 10, 10 – 50, and 50 – 1000 μm. (B) The relative abundance of the four characteristic pore diameter ranges. Bars indicate standard errors. Different letters within the same pore size range indicate statistically significant difference at p < 0.15.**



A unique observation is that the topsoil NT-CC has the widest spread of PSD, with statistically more proportion of the smaller and larger diameter pores (i.e., <0.2 µm and 50 – 1000 µm, respectively, at p < 0.15) and a bimodal distribution which is not present in the other systems (Figure 5B). Several studies have similarly observed an increase in the proportion of larger pores

in NT systems (Tavares Filho and Tessier, 2009; Pires et al., 2017; Gao et al., 2019, 2017). The reason for the abundance of small and large pores for the NT-CC systems suggests the formation of tightly packed aggregates with smaller pores and larger interaggregate pores between them. This would be consistent with results from a previous study of our site and others that found higher aggregate stability for the NT-CC systems (Mitchell et al., 2017; Gao et al., 2019). Greenland (1977) suggests soil pore size classification based on equivalent diameter into three groups as transmission (50 – 500 µm), storage (0.5 – 50

µm), and residual pores (< 0.5 µm). Larger transmission pores are important for infiltration, drainage, and aeration while smaller storage pores are important in retaining water. Increased aeration of soil is beneficial for many soil processes including healthy soil organic matter cycling (Lehmann and Kleber, 2015; Janzen, 2015) and other biogeochemical processes (Ekschmitt et al., 2008; Schmidt et al., 2011). All the treatments had higher relative abundance of the larger macropores (50 – 1000 µm) compared to ST-NO plots and ST-NO had the highest proportion (Figure 5B).

**Table 1 Modal diameter [µm] of the pore size distribution curves.**

| Depth | ST-NO | ST-CC | NT-NO | NT-CC |
|---|---|---|---|---|
| **0 - 5 cm** | 14 | 19 | 14 | 4 and 518 |
| **20 - 25 cm** | 33 | 25 | 47 | 30 |

For the subsurface soils, the combined effect of NT and CC increased the spread of PSD however, NT without CC showed a narrower PSD with the highest PSD mode and highest abundance of large macropores compared to other treatments. NT-CC plots showed a significantly higher proportion of intra-aggregate size pores and smaller (< 10 µm) at p < 0.15. Plant roots are

important actors in soil structure development, they enhance aggregation by compacting soils through growth and exudation of segmenting materials, and also fragmenting aggregates to create larger interaggregate pores (Jarvis, 2007; Angers and Caron, 1998). Given the reduced tillage in the NT plots, it could be that CC play a more critical role in forming more diverse aggregate sizes and wider PSD. The effect of the CC species should also be considered in this interpretation since it has recently been shown that the effect of CC on soil structure and porosity varies significantly with root morphology and architecture of the CC

plant (Bacq-Labreuil et al., 2019).





## 3.2 Bulk density

The mean $\rho_b$ across all treatments for the top and subsurface layer soils was 1.19 and 1.46 g cm$^{-3}$, respectively (which is equivalent to total porosities of 55 and 45 percent). Among the treatments, there was no statistically significant difference in $\rho_b$ and total porosity at $p < 0.05$ but at lower confidence levels, the topsoils under NT-NO system had significantly higher $\rho_b$

compared to ST-NO ($p = 0.078$) and NT-CC($p = 0.141$) (Figure 6 and Table B1). One of the concerns of NT practice is that it may lead to soil consolidation and an increase in compaction because of the lack of intensive tillage (Blanco-Canqui and Ruis, 2018; Moret and Arrúe, 2007). Compaction reduces soil pore volume and affects soil fertility by reducing water flow and aeration, which negatively affect soil biological activity and redox potential (Vereecken et al., 2016). Our findings show that continued long-term NT led to a slight increase in compaction, but only when practiced without CC. The changes in PSD

(discussed in section 0), however, showed that the NT systems increased the PSD in a manner that suggested a better-developed soil structure with primary and secondary structures.

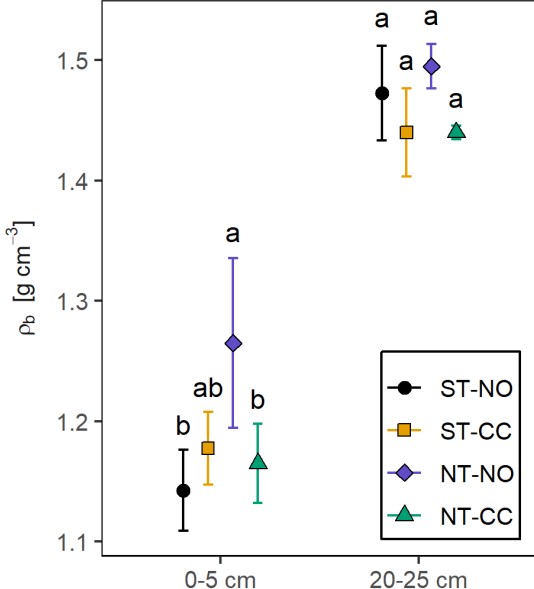

**Figure 6 Mean bulk densities ($\rho_b$) of the top (0 – 5 cm) and subsurface (20 – 25 cm) layer soils. Bars indicate standard errors. Different letters within the same depth indicate statistically significant difference at $p < 0.15$.**



### 3.3 Hydraulic conductivity

The CC treatments tended to have a greater impact on $K_s$ than the tillage treatment for the top layer soils (Figure 7). This is consistent with the increase in infiltration reported previously for our soils by Mitchell et al. (2017). They found that CC increased infiltration by 2.8 times. They suggest several possible explanations for this including increased slaking associated with ST, better formation of macropores, and better continuity of soil pores possibly due to better-established soil structure and biology (Pires et al., 2017; Schwen et al., 2011). The NT-CC systems showed higher $K_s$ than NT-NO (p = 0.011) while The ST plots showed $K_s$ midway between the NT-NO and NT-CC plots. The fact that $K_s$ of NT-NO plots is lower even more than ST plots suggests that CC is even more important when NT is practiced to maintain larger transmission pores without tillage. The effect of CC on ST plots was small and not statistically significant. $K$ at 100 cm suction, $K(100\ cm)$, is controlled by smaller pores as opposed to $K_s$. The NT plots had lower $K(100\ cm)$ compared to ST plots (Figure 7), which implies that when soils are unsaturated, the NT plots will lose water to deep drainage at a slower rate than ST plots. This could possibly mitigate the impact of reduced $\theta_{FC}$ in the NT plots and lead to an increase of water availability to plants this explanation is consistent with our results from the numerical simulation (see section 3.5).




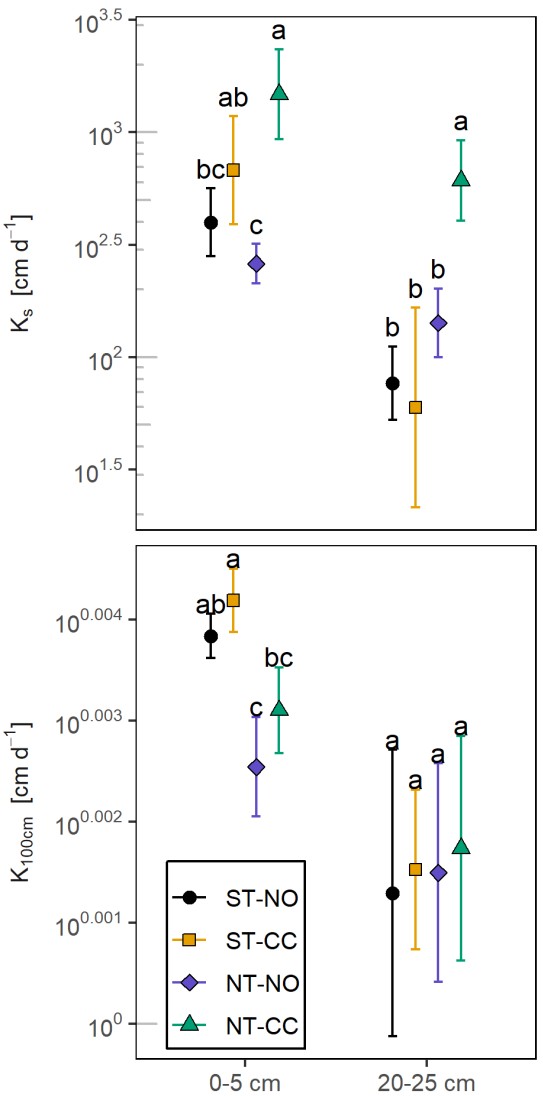

**Figure 7 Mean hydraulic conductivities at saturation ($K_s$) and 100 cm suction ($K_{100cm}$). Bars indicate standard errors. Different letters within the same depth indicate statistically significant difference at p < 0.15.**



### 3.4 Water retention

The NT treatments had lower $\theta_{FC}$ compared to ST (Figure 8). The larger value of $\theta_{FC}$ for ST plots are consistent with a more loose soil due to tillage increasing the capillary size pores. The $\theta_{FC}$ for the top layer NT soils were lower by more than 5 %

volumetric water content (p<0.016) compared to ST-CC. The ST-NO treatments had intermediate values that were not statistically different (p<0.15) from all other treatments except NT-CC. The $\theta_{FC}$ showed similar trends for the subsurface layer soils but with smaller magnitudes of differences. CC appeared to enhance the effects of NT in terms of $\theta_{FC}$ and PAW of topsoil layers (Figure 8). The NT-NO top layer soils showed values between NT-CC and the ST soils. The top layers of NT-CC plots showed a statistically significant decrease in PAW (p < 0.014) compared to the ST treatments. Assuming the top sample PAW

represents 0 – 20 cm depth and the subsurface PAW represents 20 – 40 cm depths, the NT-CC soils store 5.05 cm of equivalent surface water in plant-available form on the top 40 cm soil profile. This is 1.70 cm less plant available equivalent surface water per 40 cm depth compared to the average of the ST systems. The differences in PAW among the systems was mainly driven by $\theta_{FC}$ rather than $\theta_{PWP}$. On both layers, the CC treatment increased $\theta_{FC}$ of ST soils but had the opposite effect on the NT soils. While some studies reported an increase in $\theta_{FC}$ and PAW with CC (Basche et al., 2016b; Bilek, 2007; Villamil et al.,

2006), our findings are consistent with the observations from a recent meta-analysis of 93 paired observations of CC (Basche and DeLonge, 2017) which showed that CC did not affect total porosity for treatments practiced longer than 7 years or clay contents > 25 % which match the parameters of our study site. Our findings also agree with the findings of Basche and DeLonge (2017) in terms of $\theta_{FC}$, they find that while long-term CC tends to increase $\theta_{FC}$, it actually tends to decrease it for soils with >25% clay. Our results showed that while this was the case with ST, it was not the case for NT. For the subsurface layer of

NT treatments, $\theta_{FC}$ was significantly lower for the NT-CC compared with NT-NO treatments.





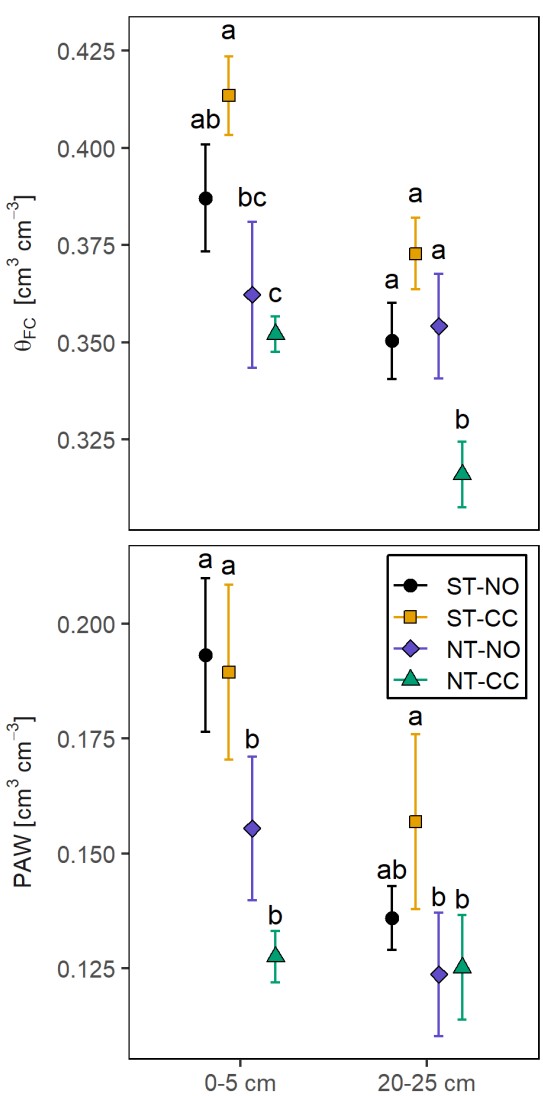

**Figure 8 Means of water content at field capacity ($\theta_{FC}$) and plant available water (PAW). Bars indicate standard errors. Different letters within the same depth indicate statistically significant difference at p < 0.15.**





## 3.5 Simulated water storage


The simulation results showed that the difference in soil water content between the different treatments is most distinct in the top 40 cm. Figure 9 shows the vertical distribution of soil moisture following the irrigation for selected times. The 2-dimension distribution of soil moisture is shown in Appendix A Figures A4 and A5. Throughout the dry down following irrigation, the CC plots maintain higher volumetric water content in the top 20 cm. In the underlying 20 – 30 cm depth layers, however, the NT-CC plots maintain the lowest soil moisture. While the NT-NO plots maintain a moderate soil water content in the top 20 cm compared to the other treatments, these plots maintain the highest water content in the 20 – 30 cm depth layers.


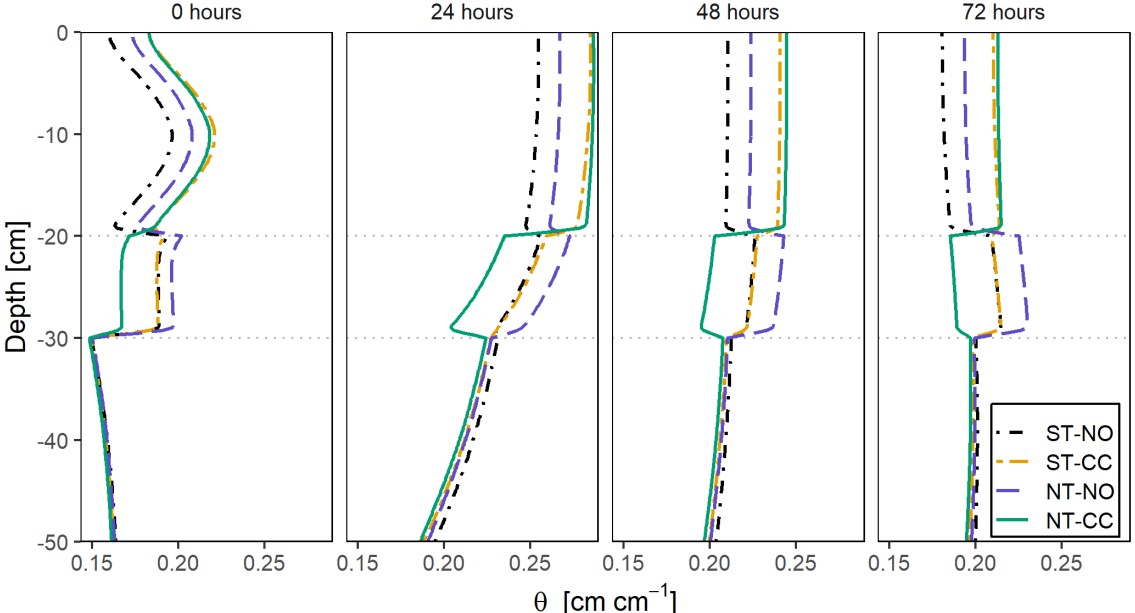

**Figure 9 Vertical soil water content distribution 0-, 24-, 48- and 72-hours after irrigation. Grey, dotted horizontal lines indicate the different soil boundaries.**


Changes in water storage over time following 4.8 cm equivalent depth irrigation (see Equation 5) are shown in Figure 10. The results show that immediately following the end of irrigation, the top 40 cm layers start to lose water (to evapotranspiration and drainage) while the deepest layer (60 – 100 cm) continues to gain water past 5 days after irrigation.



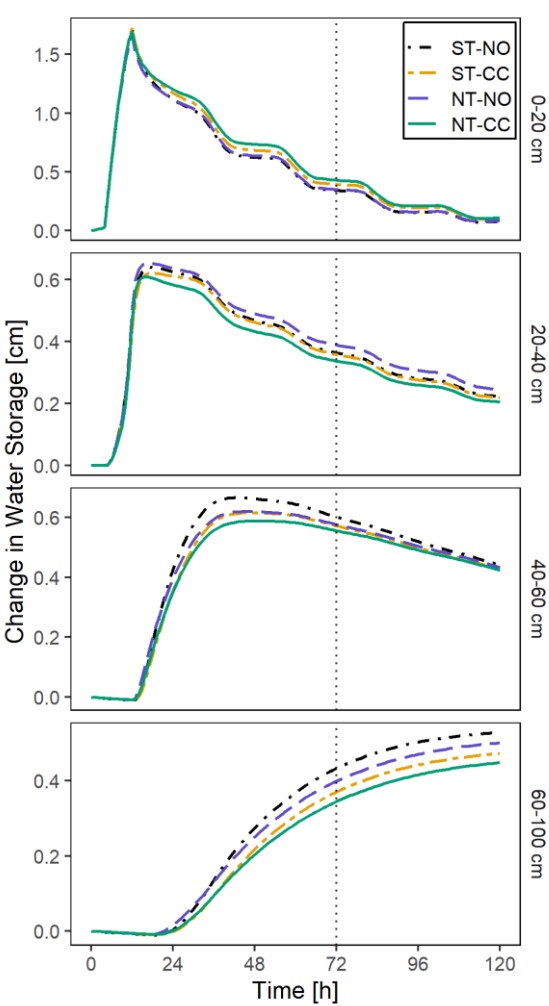

**Figure 10 Time series of changes in water storage across soil layers. Grey, dotted vertical line indicates day three after irrigation.**

The conventional measure of plant-available water storage ($PAW = \theta_{FC} - \theta_{PWP}$) relies only on the WRC. Since WRC is a description of soil water status at equilibrium, this measure of plant-available water does not account for the dynamic interactions of water retention and hydraulic conductivity (Twarakavi et al., 2009). An alternative measure of field capacity is the "dynamic field capacity" which can be defined as the amount of water maintained in the soil after excess gravitational water is drained and the rate of downward movement is minimal (Veihmeyer and Hendrickson, 1931). This dynamic field





capacity is commonly taken as the water content after three (or sometimes even five) days (Twarakavi et al., 2009; Assouline and Or, 2014). In our simulation, the rate of water drainage for the top and middle layers had significantly decreased after three days (Figure 10).

Comparison of the treatment averages in volumetric water content and amount of water retained three days after irrigation (that is the dynamic field capacity and water storage at time of field capacity) are shown in Figure 11. The magnitude of differences among all treatments were small but generally favored the NT and CC treatments. For the top 20 cm soils, the only statistically significant difference in change in water storage was between NT-CC and ST-NO plots (p = 0.12) with both the ST-CC and NT-NO showing intermediate storage between the two. For the top 20 cm soils, the water content at dynamic field

capacity for the CC plots was higher than those for NO plots, with the ST-NO plots showing statistically significant (p < 0.09) lower water content than both CC plots. At the 20 – 40 cm depths, the only statistically significant difference is between NT-NO and NT-CC with the NT-NO plots holding the most amount of water while the NT-CC holds the least amount. Both the ST plots, with and without CC show no difference in water content or water storage change three days after irrigation. These findings of water content at field capacity contrast with the $\theta_{FC}$ the PAW estimated from the conventional steady-state

measures (see Figure 8) which showed that the ST plots, in general, had higher water contents at field capacity and higher PAW. The dynamic water content at field capacity for the subsurface layers 20 – 30 cm shows similarity with that of the conventional steady-state field capacity for 20 -25 cm soils in that the NT-CC plots have lower water contents compared to NT-NO (p < 0.06). The ST plots 20 – 40 cm have water content at dynamic field capacity closer to that of NT-NO but not statistically different from that of NT-CC (at P < 0.15). Only the dynamic water storage and water contents at field capacity

capture the interaction between water retention curve and hydraulic conductivity functions, therefore these measures likely capture soil hydrology more accurately.

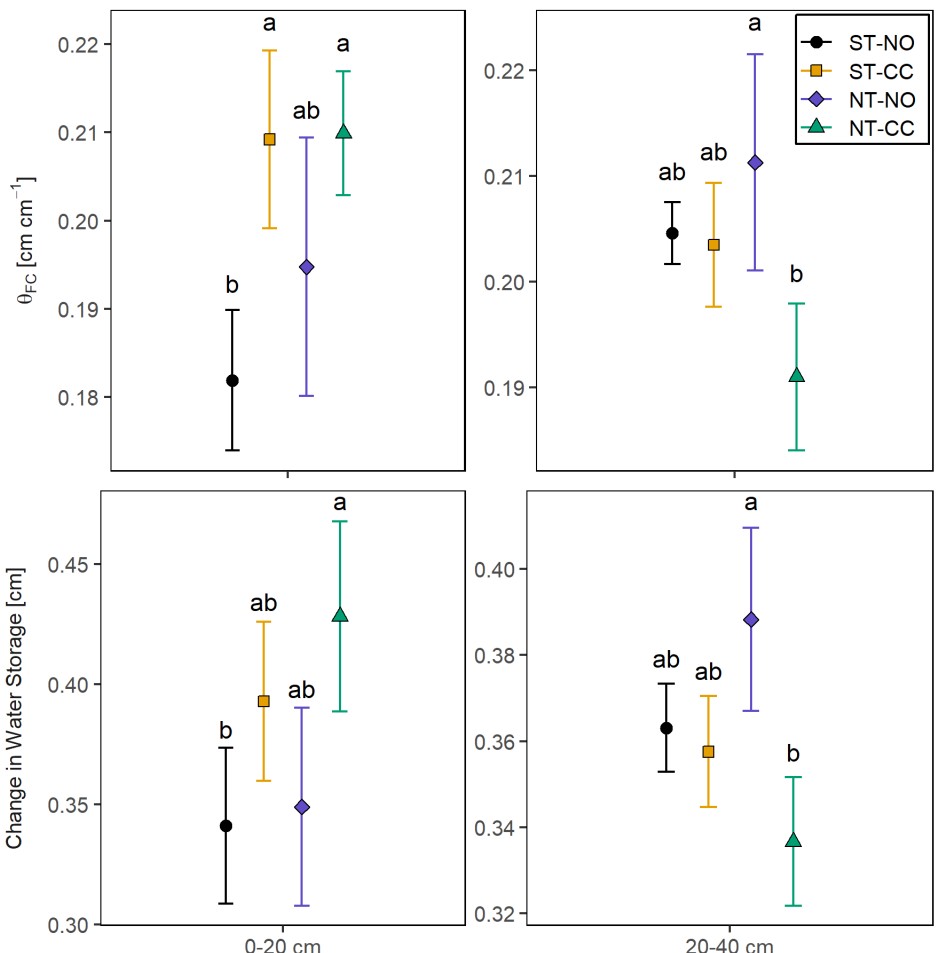

**Figure 11 Dynamic field capacity ($\theta_{FC}$) and water storage change day three after irrigation. Bars indicate standard errors. Different letters indicate statistically significant difference at p < 0.15.**


## 4 Conclusion

The long-term NT and CC practices had an impact on soil pore size distribution (PSD). The NT and CC practiced independently led to a moderate increase in PSD range and had small or no effect on the measured soil hydraulic properties and simulated water dynamics. On the plots where NT and CC are practiced together, the changes in soil structure and hydraulic properties were most pronounced. NT with CC led to the development of bimodal pore size distribution in the top (0 –5 cm) soils with






the modes of the PSD around 4 and 500 μm diameter sizes which are in the storage and transmission pore size categories, respectively. While ST is done to improve soil structure for crops and overcome the compaction of the topsoil layer, its effect is transitory. Our results suggest that in the longer term, NT and CC increase soil aggregation and the proportion of larger pores while also maintaining total porosity.

CC practices increased the saturated hydraulic conductivity ($K_s$), particularly when practiced in conjunction with NT practices. For the top layer soils (0 – 5 cm), the $K_s$ of the NT-CC soils was significantly higher (p = 0.01) than that of the NT-NO soils. The $K_s$ of NT-CC subsurface layer (20 – 25 cm) was significantly higher (p < 0.15) than all other systems.

The measured water retention suggested a decrease in soils' ability to store water. The NT-CC practices decreased the calculated plant-available water (PAW) and water content at field capacity ($\theta_{FC}$). While these steady-state measures of field

capacity and PAW indicate soil's ability to store water, the dynamically simulated water storage in soils is the result of the interaction between soil's water retention characteristics and its hydraulic conductivities. Both the water retention and conductivity were accounted for in the HYDRUS-2D irrigation simulation. The results showed that when both retention and conductivity properties are considered together in the simulation, the top layers of NT systems not only do not have a disadvantage but have a marginally increased ability to store water compared to ST plots.

The changes in PSD, water conductivity, and water storage associated with NT and CC systems observed in this study suggest that these systems are beneficial to the general soil health water retention at the plot scale.



**Appendix A: Individual samples measurement curves and supplemental figures**

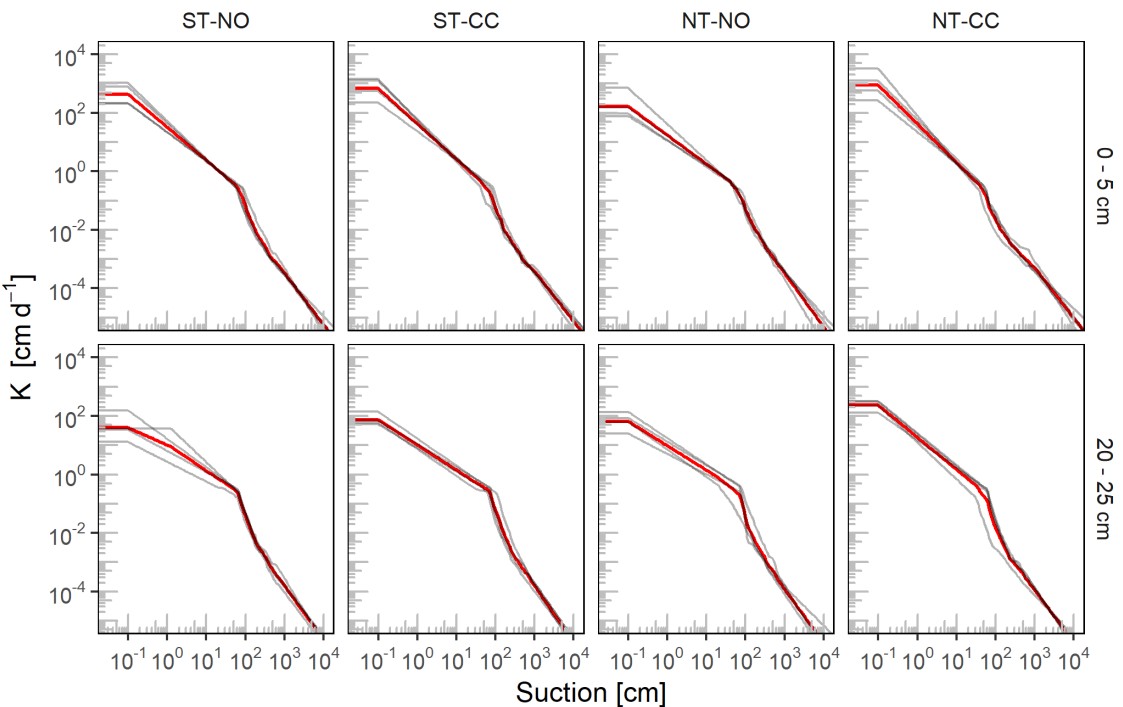

**Figure A1 Hydraulic conductivity functions of top and subsurface layers by treatment. Grey curves are individual soil core measurements and thick red curves are the treatment means.**



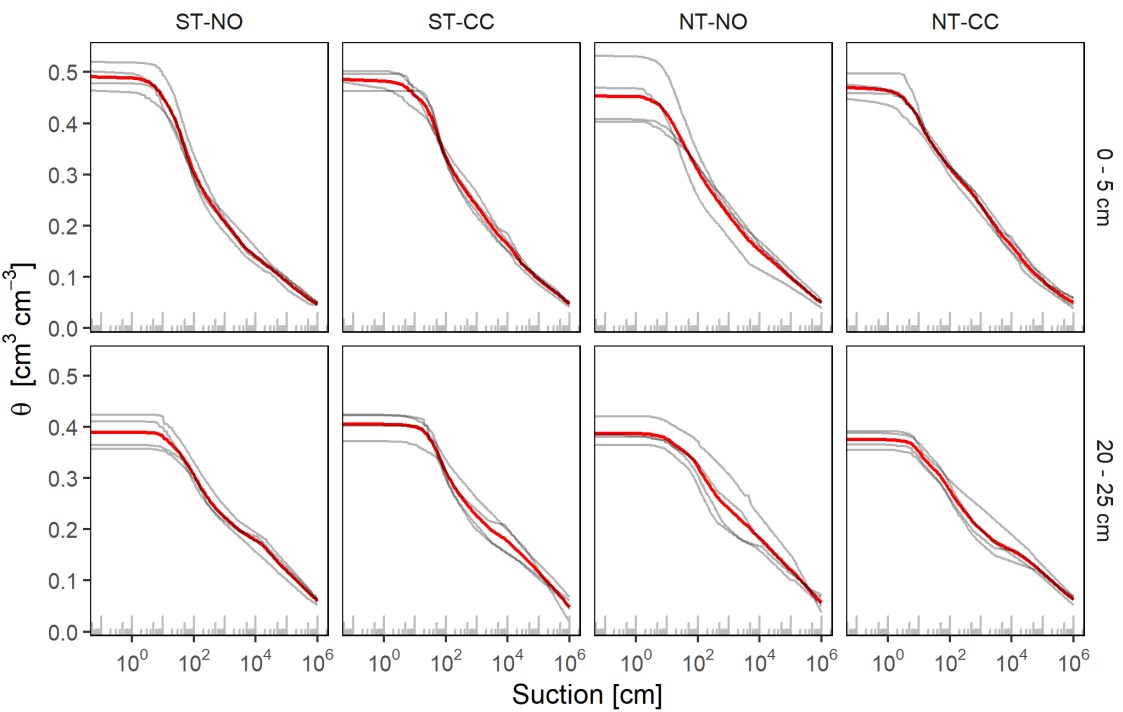

**Figure A2 Water retention curves of top and subsurface layers by treatment. Grey curves are individual soil core measurements**
**and thick red curves are the treatment means.**




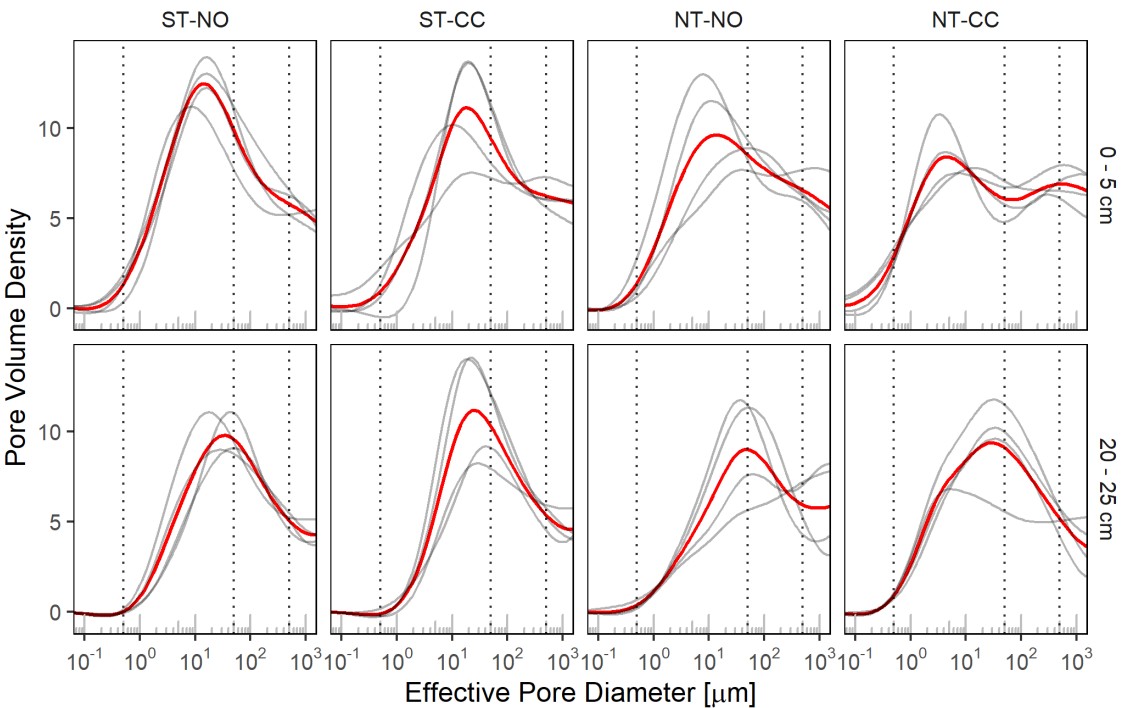

**Figure A3 Effective pore size distribution. Grey curves are individual soil core measurements and thick red curves are means of the replicates. Vertical dotted lines indicate pore diameter sizes of 0.5, 50, and 500 µm.**




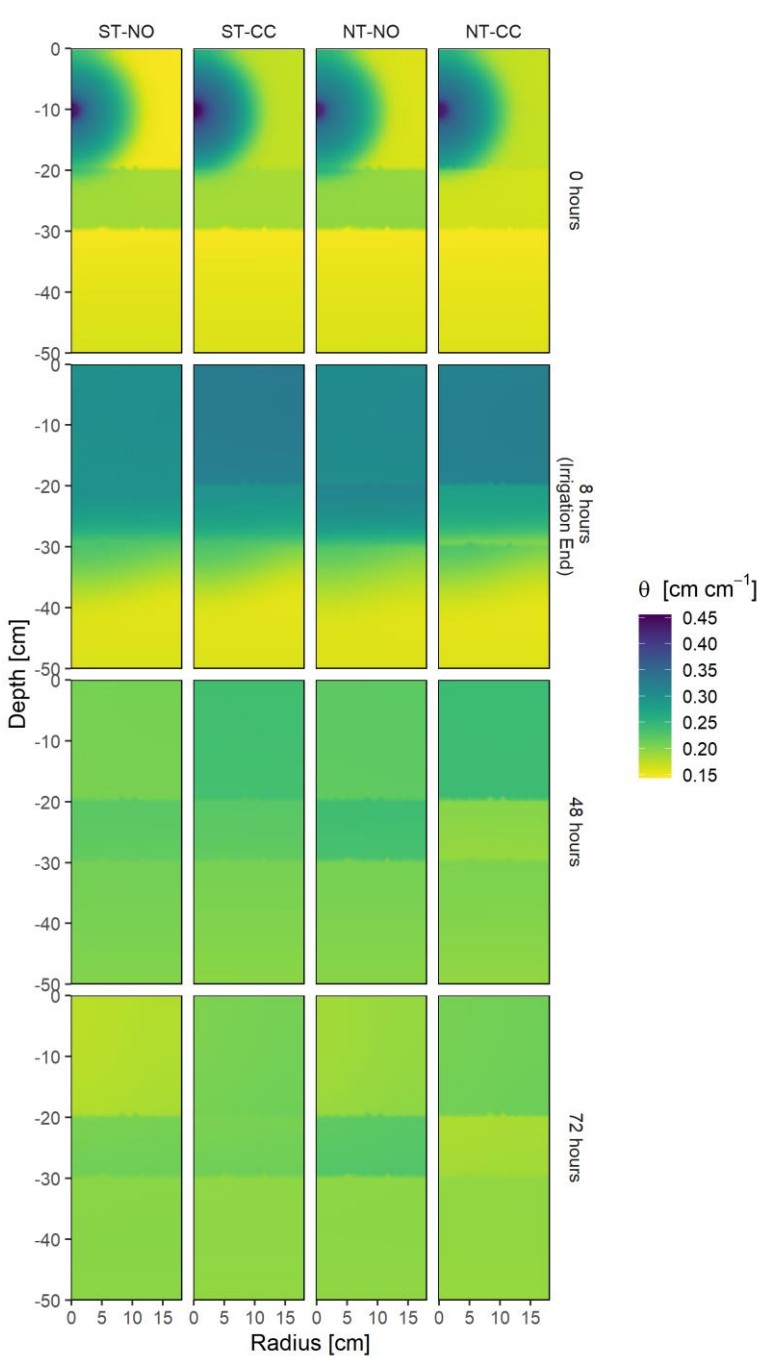

**Figure A4 Soil water content distribution in the model domain at the start of irrigation and 0-, 48-, and 72-hours after irrigation.**



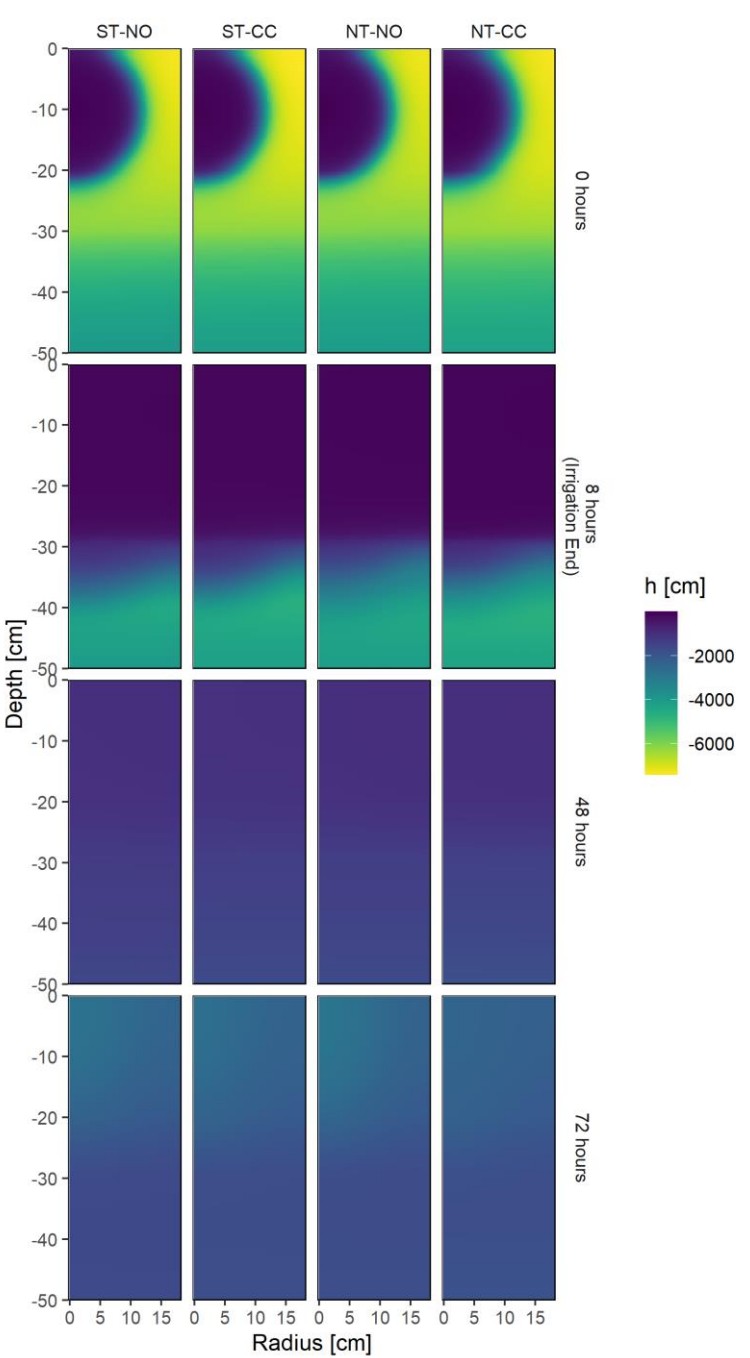

**Figure A5 Hydraulic head distribution in the model domain at the start of irrigation and 0-, 48-, and 72-hours after irrigation.**





**Appendix B: Statistical comparison of treatments**

**Table B1: Tukey's HSD test comparison of means for soil hydraulic properties. Tukey's HSD comparison of means. P-values < 0.15 are printed in bold and p-values < 0.05 bold and underlined. LCL and UCL are lower and upper control intervals, respectively.**

| Variable [unit] | Depth Range [cm] | Comparison | Difference | P-value | LCL | UCL |
|---|---|---|---|---|---|---|
| $\rho_b$ [g cm$^{-3}$] | 0-20 | NT-CC -- NT-NO | -0.1 | **0.1412** | -0.19765 | -0.00235 |
| $\rho_b$ [g cm$^{-3}$] | 0-20 | NT-CC -- ST-CC | -0.0125 | 0.8472 | -0.11015 | 0.085148 |
| $\rho_b$ [g cm$^{-3}$] | 0-20 | NT-CC -- ST-NO | 0.0225 | 0.7292 | -0.07515 | 0.120148 |
| $\rho_b$ [g cm$^{-3}$] | 0-20 | NT-NO -- ST-CC | 0.0875 | 0.1933 | -0.01015 | 0.185148 |
| $\rho_b$ [g cm$^{-3}$] | 0-20 | NT-NO -- ST-NO | 0.1225 | **0.0777** | 0.024852 | 0.220148 |
| $\rho_b$ [g cm$^{-3}$] | 0-20 | ST-CC -- ST-NO | 0.035 | 0.5916 | -0.06265 | 0.132648 |
| $\rho_b$ [g cm$^{-3}$] | 20-25 | NT-CC -- NT-NO | -0.055 | 0.1984 | -0.11714 | 0.007136 |
| $\rho_b$ [g cm$^{-3}$] | 20-25 | NT-CC -- ST-CC | 0 | 1 | -0.06214 | 0.062136 |
| $\rho_b$ [g cm$^{-3}$] | 20-25 | NT-CC -- ST-NO | -0.0325 | 0.4368 | -0.09464 | 0.029636 |
| $\rho_b$ [g cm$^{-3}$] | 20-25 | NT-NO -- ST-CC | 0.055 | 0.1984 | -0.00714 | 0.117136 |
| $\rho_b$ [g cm$^{-3}$] | 20-25 | NT-NO -- ST-NO | 0.0225 | 0.5878 | -0.03964 | 0.084636 |
| $\rho_b$ [g cm$^{-3}$] | 20-25 | ST-CC -- ST-NO | -0.0325 | 0.4368 | -0.09464 | 0.029636 |
| $\theta_{-33kPa}$ [cm$^3$ cm$^{-3}$] | 0-20 | NT-CC -- NT-NO | -0.01017 | 0.5868 | -0.03817 | 0.017834 |
| $\theta_{-33kPa}$ [cm$^3$ cm$^{-3}$] | 0-20 | NT-CC -- ST-CC | -0.06136 | **0.0056** | -0.08936 | -0.03336 |
| $\theta_{-33kPa}$ [cm$^3$ cm$^{-3}$] | 0-20 | NT-CC -- ST-NO | -0.03503 | **0.0784** | -0.06303 | -0.00703 |
| $\theta_{-33kPa}$ [cm$^3$ cm$^{-3}$] | 0-20 | NT-NO -- ST-CC | -0.05119 | **0.0157** | -0.07919 | -0.02319 |
| $\theta_{-33kPa}$ [cm$^3$ cm$^{-3}$] | 0-20 | NT-NO -- ST-NO | -0.02487 | 0.1971 | -0.05287 | 0.003135 |
| $\theta_{-33kPa}$ [cm$^3$ cm$^{-3}$] | 0-20 | ST-CC -- ST-NO | 0.026326 | 0.1738 | -0.00167 | 0.054328 |
| $\theta_{-33kPa}$ [cm$^3$ cm$^{-3}$] | 20-25 | NT-CC -- NT-NO | -0.03819 | **0.0234** | -0.06082 | -0.01556 |
| $\theta_{-33kPa}$ [cm$^3$ cm$^{-3}$] | 20-25 | NT-CC -- ST-CC | -0.05679 | **0.0023** | -0.07942 | -0.03417 |
| $\theta_{-33kPa}$ [cm$^3$ cm$^{-3}$] | 20-25 | NT-CC -- ST-NO | -0.03436 | **0.0377** | -0.05698 | -0.01173 |
| $\theta_{-33kPa}$ [cm$^3$ cm$^{-3}$] | 20-25 | NT-NO -- ST-CC | -0.0186 | 0.2301 | -0.04123 | 0.004023 |
| $\theta_{-33kPa}$ [cm$^3$ cm$^{-3}$] | 20-25 | NT-NO -- ST-NO | 0.003836 | 0.7987 | -0.01879 | 0.026462 |
| $\theta_{-33kPa}$ [cm$^3$ cm$^{-3}$] | 20-25 | ST-CC -- ST-NO | 0.022439 | 0.1531 | -0.00019 | 0.045066 |
| $\theta_{-10kPa}$ [cm$^3$ cm$^{-3}$] | 0-20 | NT-CC -- NT-NO | -0.01017 | 0.6712 | -0.04613 | 0.025786 |
| $\theta_{-10kPa}$ [cm$^3$ cm$^{-3}$] | 0-20 | NT-CC -- ST-CC | -0.04686 | **0.0682** | -0.08282 | -0.0109 |
| $\theta_{-10kPa}$ [cm$^3$ cm$^{-3}$] | 0-20 | NT-CC -- ST-NO | -0.04269 | **0.0929** | -0.07865 | -0.00673 |
| $\theta_{-10kPa}$ [cm$^3$ cm$^{-3}$] | 0-20 | NT-NO -- ST-CC | -0.03668 | **0.1427** | -0.07264 | -0.00072 |
| $\theta_{-10kPa}$ [cm$^3$ cm$^{-3}$] | 0-20 | NT-NO -- ST-NO | -0.03251 | 0.1896 | -0.06847 | 0.003448 |
| $\theta_{-10kPa}$ [cm$^3$ cm$^{-3}$] | 0-20 | ST-CC -- ST-NO | 0.00417 | 0.8614 | -0.03179 | 0.04013 |
| $\theta_{-10kPa}$ [cm$^3$ cm$^{-3}$] | 20-25 | NT-CC -- NT-NO | -0.0236 | 0.2088 | -0.05092 | 0.003728 |
| $\theta_{-10kPa}$ [cm$^3$ cm$^{-3}$] | 20-25 | NT-CC -- ST-CC | -0.04766 | **0.0199** | -0.07498 | -0.02033 |
| $\theta_{-10kPa}$ [cm$^3$ cm$^{-3}$] | 20-25 | NT-CC -- ST-NO | -0.03037 | **0.1131** | -0.05769 | -0.00304 |
| $\theta_{-10kPa}$ [cm$^3$ cm$^{-3}$] | 20-25 | NT-NO -- ST-CC | -0.02406 | 0.2006 | -0.05139 | 0.003263 |
| $\theta_{-10kPa}$ [cm$^3$ cm$^{-3}$] | 20-25 | NT-NO -- ST-NO | -0.00677 | 0.7099 | -0.03409 | 0.020556 |
| $\theta_{-10kPa}$ [cm$^3$ cm$^{-3}$] | 20-25 | ST-CC -- ST-NO | 0.017293 | 0.3496 | -0.01003 | 0.044618 |
| $K_{-10kPa}$ [log$_{10}$(cm d$^{-1}$)] | 0-20 | NT-CC -- NT-NO | 0.00056 | 0.3149 | -0.00026 | 0.00138 |
| $K_{-10kPa}$ [log$_{10}$(cm d$^{-1}$)] | 0-20 | NT-CC -- ST-CC | -0.00109 | **0.0642** | -0.00191 | -0.00027 |
| $K_{-10kPa}$ [log$_{10}$(cm d$^{-1}$)] | 0-20 | NT-CC -- ST-NO | -0.00074 | 0.1933 | -0.00156 | 8.53E-05 |
| $K_{-10kPa}$ [log$_{10}$(cm d$^{-1}$)] | 0-20 | NT-NO -- ST-CC | -0.00165 | **0.0094** | -0.00247 | -0.00083 |
| $K_{-10kPa}$ [log$_{10}$(cm d$^{-1}$)] | 0-20 | NT-NO -- ST-NO | -0.00129 | **0.0319** | -0.00212 | -0.00047 |
| $K_{-10kPa}$ [log$_{10}$(cm d$^{-1}$)] | 0-20 | ST-CC -- ST-NO | 0.000352 | 0.5217 | -0.00047 | 0.001173 |
| $K_{-10kPa}$ [log$_{10}$(cm d$^{-1}$)] | 20-25 | NT-CC -- NT-NO | 0.000242 | 0.8814 | -0.0022 | 0.002682 |



| Variable [unit] | Depth Range [cm] | Comparison | Difference | P-value | LCL | UCL |
|---|---|---|---|---|---|---|
| $K_{-10kPa}$ [$\log_{10}$(cm d$^{-1}$)] | 20-25 | NT-CC -- ST-CC | 0.000211 | 0.8966 | -0.00223 | 0.002651 |
| $K_{-10kPa}$ [$\log_{10}$(cm d$^{-1}$)] | 20-25 | NT-CC -- ST-NO | 0.000443 | 0.7848 | -0.002 | 0.002883 |
| $K_{-10kPa}$ [$\log_{10}$(cm d$^{-1}$)] | 20-25 | NT-NO -- ST-CC | -3.1E-05 | 0.9847 | -0.00247 | 0.002409 |
| $K_{-10kPa}$ [$\log_{10}$(cm d$^{-1}$)] | 20-25 | NT-NO -- ST-NO | 0.000201 | 0.9012 | -0.00224 | 0.002642 |
| $K_{-10kPa}$ [$\log_{10}$(cm d$^{-1}$)] | 20-25 | ST-CC -- ST-NO | 0.000232 | 0.886 | -0.00221 | 0.002673 |
| Ks [$\log_{10}$(cm d$^{-1}$)] | 0-20 | NT-CC -- NT-NO | 0.75351 | **0.0116** | 0.363698 | 1.143322 |
| Ks [$\log_{10}$(cm d$^{-1}$)] | 0-20 | NT-CC -- ST-CC | 0.337974 | 0.2071 | -0.05184 | 0.727786 |
| Ks [$\log_{10}$(cm d$^{-1}$)] | 0-20 | NT-CC -- ST-NO | 0.570122 | **0.044** | 0.18031 | 0.959934 |
| Ks [$\log_{10}$(cm d$^{-1}$)] | 0-20 | NT-NO -- ST-CC | -0.41554 | **0.1271** | -0.80535 | -0.02572 |
| Ks [$\log_{10}$(cm d$^{-1}$)] | 0-20 | NT-NO -- ST-NO | -0.18339 | 0.4832 | -0.5732 | 0.206424 |
| Ks [$\log_{10}$(cm d$^{-1}$)] | 0-20 | ST-CC -- ST-NO | 0.232148 | 0.3778 | -0.15766 | 0.62196 |
| Ks [$\log_{10}$(cm d$^{-1}$)] | 20-25 | NT-CC -- NT-NO | 0.633404 | **0.1155** | 0.059248 | 1.20756 |
| Ks [$\log_{10}$(cm d$^{-1}$)] | 20-25 | NT-CC -- ST-CC | 1.009435 | **0.0192** | 0.435279 | 1.583591 |
| Ks [$\log_{10}$(cm d$^{-1}$)] | 20-25 | NT-CC -- ST-NO | 0.900776 | **0.0327** | 0.32662 | 1.474932 |
| Ks [$\log_{10}$(cm d$^{-1}$)] | 20-25 | NT-NO -- ST-CC | 0.376031 | 0.3337 | -0.19813 | 0.950187 |
| Ks [$\log_{10}$(cm d$^{-1}$)] | 20-25 | NT-NO -- ST-NO | 0.267372 | 0.4876 | -0.30678 | 0.841528 |
| Ks [$\log_{10}$(cm d$^{-1}$)] | 20-25 | ST-CC -- ST-NO | -0.10866 | 0.776 | -0.68282 | 0.465497 |
| PAW [cm$^3$ cm$^{-3}$] | 0-20 | NT-CC -- NT-NO | -0.02791 | 0.2175 | -0.06088 | 0.005069 |
| PAW [cm$^3$ cm$^{-3}$] | 0-20 | NT-CC -- ST-CC | -0.06189 | **0.0137** | -0.09486 | -0.02891 |
| PAW [cm$^3$ cm$^{-3}$] | 0-20 | NT-CC -- ST-NO | -0.06563 | **0.0099** | -0.0986 | -0.03265 |
| PAW [cm$^3$ cm$^{-3}$] | 0-20 | NT-NO -- ST-CC | -0.03398 | **0.139** | -0.06696 | -0.00101 |
| PAW [cm$^3$ cm$^{-3}$] | 0-20 | NT-NO -- ST-NO | -0.03772 | **0.104** | -0.0707 | -0.00474 |
| PAW [cm$^3$ cm$^{-3}$] | 0-20 | ST-CC -- ST-NO | -0.00374 | 0.8645 | -0.03671 | 0.029238 |
| PAW [cm$^3$ cm$^{-3}$] | 20-25 | NT-CC -- NT-NO | 0.001511 | 0.9378 | -0.02765 | 0.03067 |
| PAW [cm$^3$ cm$^{-3}$] | 20-25 | NT-CC -- ST-CC | -0.03174 | **0.12** | -0.0609 | -0.00258 |
| PAW [cm$^3$ cm$^{-3}$] | 20-25 | NT-CC -- ST-NO | -0.01075 | 0.5812 | -0.03991 | 0.018411 |
| PAW [cm$^3$ cm$^{-3}$] | 20-25 | NT-NO -- ST-CC | -0.03325 | **0.1049** | -0.06241 | -0.00409 |
| PAW [cm$^3$ cm$^{-3}$] | 20-25 | NT-NO -- ST-NO | -0.01226 | 0.5301 | -0.04142 | 0.0169 |
| PAW [cm$^3$ cm$^{-3}$] | 20-25 | ST-CC -- ST-NO | 0.020992 | 0.2899 | -0.00817 | 0.050151 |
| $\phi$ [cm$^3$ cm$^{-3}$] | 0-20 | NT-CC -- NT-NO | 0.0375 | 0.1642 | -0.00143 | 0.076427 |
| $\phi$ [cm$^3$ cm$^{-3}$] | 0-20 | NT-CC -- ST-CC | 0.005 | 0.8467 | -0.03393 | 0.043927 |
| $\phi$ [cm$^3$ cm$^{-3}$] | 0-20 | NT-CC -- ST-NO | -0.01 | 0.6997 | -0.04893 | 0.028927 |
| $\phi$ [cm$^3$ cm$^{-3}$] | 0-20 | NT-NO -- ST-CC | -0.0325 | 0.2234 | -0.07143 | 0.006427 |
| $\phi$ [cm$^3$ cm$^{-3}$] | 0-20 | NT-NO -- ST-NO | -0.0475 | **0.0851** | -0.08643 | -0.00857 |
| $\phi$ [cm$^3$ cm$^{-3}$] | 0-20 | ST-CC -- ST-NO | -0.015 | 0.5644 | -0.05393 | 0.023927 |
| $\phi$ [cm$^3$ cm$^{-3}$] | 20-25 | NT-CC -- NT-NO | 0.02 | 0.214 | -0.00344 | 0.04344 |
| $\phi$ [cm$^3$ cm$^{-3}$] | 20-25 | NT-CC -- ST-CC | -0.0025 | 0.8724 | -0.02594 | 0.02094 |
| $\phi$ [cm$^3$ cm$^{-3}$] | 20-25 | NT-CC -- ST-NO | 0.01 | 0.5241 | -0.01344 | 0.03344 |
| $\phi$ [cm$^3$ cm$^{-3}$] | 20-25 | NT-NO -- ST-CC | -0.0225 | 0.1656 | -0.04594 | 0.00094 |
| $\phi$ [cm$^3$ cm$^{-3}$] | 20-25 | NT-NO -- ST-NO | -0.01 | 0.5241 | -0.03344 | 0.01344 |
| $\phi$ [cm$^3$ cm$^{-3}$] | 20-25 | ST-CC -- ST-NO | 0.0125 | 0.4281 | -0.01094 | 0.03594 |
| $\theta_{-1500kPa}$ [cm$^3$ cm$^{-3}$] | 0-20 | NT-CC -- NT-NO | 0.01774 | 0.2361 | -0.00414 | 0.039617 |
| $\theta_{-1500kPa}$ [cm$^3$ cm$^{-3}$] | 0-20 | NT-CC -- ST-CC | 0.000529 | 0.9709 | -0.02135 | 0.022406 |
| $\theta_{-1500kPa}$ [cm$^3$ cm$^{-3}$] | 0-20 | NT-CC -- ST-NO | 0.030594 | **0.0526** | 0.008716 | 0.052471 |
| $\theta_{-1500kPa}$ [cm$^3$ cm$^{-3}$] | 0-20 | NT-NO -- ST-CC | -0.01721 | 0.2496 | -0.03909 | 0.004666 |
| $\theta_{-1500kPa}$ [cm$^3$ cm$^{-3}$] | 0-20 | NT-NO -- ST-NO | 0.012853 | 0.384 | -0.00902 | 0.034731 |
| $\theta_{-1500kPa}$ [cm$^3$ cm$^{-3}$] | 0-20 | ST-CC -- ST-NO | 0.030064 | **0.0562** | 0.008187 | 0.051942 |
| $\theta_{-1500kPa}$ [cm$^3$ cm$^{-3}$] | 20-25 | NT-CC -- NT-NO | -0.0397 | **0.1168** | -0.07583 | -0.00357 |



| Variable [unit] | Depth Range [cm] | Comparison | Difference | P-value | LCL | UCL |
|---|---|---|---|---|---|---|
| $\theta_{-1500kPa}$ [cm$^3$ cm$^{-3}$] | 20-25 | NT-CC -- ST-CC | -0.02505 | 0.3072 | -0.06119 | 0.011078 |
| $\theta_{-1500kPa}$ [cm$^3$ cm$^{-3}$] | 20-25 | NT-CC -- ST-NO | -0.02361 | 0.3348 | -0.05974 | 0.012526 |
| $\theta_{-1500kPa}$ [cm$^3$ cm$^{-3}$] | 20-25 | NT-NO -- ST-CC | 0.014648 | 0.5446 | -0.02148 | 0.05078 |
| $\theta_{-1500kPa}$ [cm$^3$ cm$^{-3}$] | 20-25 | NT-NO -- ST-NO | 0.016095 | 0.5063 | -0.02004 | 0.052228 |
| $\theta_{-1500kPa}$ [cm$^3$ cm$^{-3}$] | 20-25 | ST-CC -- ST-NO | 0.001447 | 0.9519 | -0.03469 | 0.037579 |
| $\theta_{FC(3day)}$ [cm cm$^{-1}$] | 0 – 20 | NT-CC - NT-NO | 0.015140174 | 0.3214 | -0.00737 | 0.037654 |
| $\theta_{FC(3day)}$ [cm cm$^{-1}$] | 0 – 20 | NT-CC - ST-CC | 7.06E-04 | 0.9623 | -0.02181 | 0.02322 |
| $\theta_{FC(3day)}$ [cm cm$^{-1}$] | 0 – 20 | NT-CC - ST-NO | 0.027982868 | **0.0801** | 0.005469 | 0.050497 |
| $\theta_{FC(3day)}$ [cm cm$^{-1}$] | 0 - 20 | NT-NO - ST-CC | -0.014434127 | 0.3436 | -0.03695 | 0.00808 |
| $\theta_{FC(3day)}$ [cm cm$^{-1}$] | 0 - 20 | NT-NO - ST-NO | 0.012842694 | 0.3976 | -0.00967 | 0.035357 |
| $\theta_{FC(3day)}$ [cm cm$^{-1}$] | 0 - 20 | ST-CC - ST-NO | 0.02727682 | **0.0871** | 0.004763 | 0.049791 |
| $\theta_{FC(3day)}$ [cm cm$^{-1}$] | 20 - 40 | NT-CC - NT-NO | -0.020299355 | **0.0625** | -0.03551 | -0.00509 |
| $\theta_{FC(3day)}$ [cm cm$^{-1}$] | 20 - 40 | NT-CC - ST-CC | -0.012484612 | 0.2307 | -0.02769 | 0.002722 |
| $\theta_{FC(3day)}$ [cm cm$^{-1}$] | 20 - 40 | NT-CC - ST-NO | -0.01359903 | 0.1942 | -0.02881 | 0.001608 |
| $\theta_{FC(3day)}$ [cm cm$^{-1}$] | 20 - 40 | NT-NO - ST-CC | 0.007814743 | 0.4447 | -0.00739 | 0.023022 |
| $\theta_{FC(3day)}$ [cm cm$^{-1}$] | 20 - 40 | NT-NO - ST-NO | 0.006700325 | 0.5109 | -0.00851 | 0.021907 |
| $\theta_{FC(3day)}$ [cm cm$^{-1}$] | 20 - 40 | ST-CC - ST-NO | -0.001114417 | 0.9121 | -0.01632 | 0.014093 |
| $\theta_{FC(3day)}$ [cm cm$^{-1}$] | 40 - 60 | NT-CC - NT-NO | -0.00296152 | 0.5114 | -0.00969 | 0.003768 |
| $\theta_{FC(3day)}$ [cm cm$^{-1}$] | 40 - 60 | NT-CC - ST-CC | -0.002087109 | 0.6419 | -0.00882 | 0.004642 |
| $\theta_{FC(3day)}$ [cm cm$^{-1}$] | 40 - 60 | NT-CC - ST-NO | -0.005187568 | 0.2587 | -0.01192 | 0.001542 |
| $\theta_{FC(3day)}$ [cm cm$^{-1}$] | 40 - 60 | NT-NO - ST-CC | 8.74E-04 | 0.845 | -0.00586 | 0.007604 |
| $\theta_{FC(3day)}$ [cm cm$^{-1}$] | 40 - 60 | NT-NO - ST-NO | -0.002226048 | 0.6202 | -0.00896 | 0.004503 |
| $\theta_{FC(3day)}$ [cm cm$^{-1}$] | 40 - 60 | ST-CC - ST-NO | -0.003100459 | 0.4921 | -0.00983 | 0.003629 |
| $\theta_{FC(3day)}$ [cm cm$^{-1}$] | 60 - 100 | NT-CC - NT-NO | -0.002812337 | 0.4919 | -0.00891 | 0.003289 |
| $\theta_{FC(3day)}$ [cm cm$^{-1}$] | 60 - 100 | NT-CC - ST-CC | -0.001599995 | 0.6938 | -0.0077 | 0.004501 |
| $\theta_{FC(3day)}$ [cm cm$^{-1}$] | 60 - 100 | NT-CC - ST-NO | -0.004512793 | 0.2775 | -0.01061 | 0.001589 |
| $\theta_{FC(3day)}$ [cm cm$^{-1}$] | 60 - 100 | NT-NO - ST-CC | 0.001212341 | 0.7652 | -0.00489 | 0.007314 |
| $\theta_{FC(3day)}$ [cm cm$^{-1}$] | 60 - 100 | NT-NO - ST-NO | -0.001700456 | 0.6758 | -0.0078 | 0.004401 |
| $\theta_{FC(3day)}$ [cm cm$^{-1}$] | 60 - 100 | ST-CC - ST-NO | -0.002912798 | 0.4769 | -0.00901 | 0.003189 |
| $\Delta W_{(3 day)}$ [cm] | 0 - 20 | NT-CC - NT-NO | 0.079245899 | 0.1537 | -7.80E-04 | 0.159272 |
| $\Delta W_{(3 day)}$ [cm] | 0 - 20 | NT-CC - ST-CC | 0.035343204 | 0.5099 | -0.04468 | 0.115369 |
| $\Delta W_{(3 day)}$ [cm] | 0 - 20 | NT-CC - ST-NO | 0.087166804 | **0.1197** | 0.007141 | 0.167193 |
| $\Delta W_{(3 day)}$ [cm] | 0 - 20 | NT-NO - ST-CC | -0.043902695 | 0.4153 | -0.12393 | 0.036123 |
| $\Delta W_{(3 day)}$ [cm] | 0 - 20 | NT-NO - ST-NO | 0.007920906 | 0.8815 | -0.07211 | 0.087947 |
| $\Delta W_{(3 day)}$ [cm] | 0 - 20 | ST-CC - ST-NO | 0.0518236 | 0.3389 | -0.0282 | 0.13185 |
| $\Delta W_{(3 day)}$ [cm] | 20 - 40 | NT-CC - NT-NO | -0.051541149 | **_0.0353_** | -0.08496 | -0.01812 |
| $\Delta W_{(3 day)}$ [cm] | 20 - 40 | NT-CC - ST-CC | -0.020864525 | 0.3559 | -0.05429 | 0.012557 |
| $\Delta W_{(3 day)}$ [cm] | 20 - 40 | NT-CC - ST-NO | -0.026390739 | 0.2479 | -0.05981 | 0.007031 |
| $\Delta W_{(3 day)}$ [cm] | 20 - 40 | NT-NO - ST-CC | 0.030676624 | 0.1835 | -0.00274 | 0.064098 |
| $\Delta W_{(3 day)}$ [cm] | 20 - 40 | NT-NO - ST-NO | 0.025150411 | 0.2696 | -0.00827 | 0.058572 |
| $\Delta W_{(3 day)}$ [cm] | 20 - 40 | ST-CC - ST-NO | -0.005526213 | 0.8036 | -0.03895 | 0.027895 |
| $\Delta W_{(3 day)}$ [cm] | 40 - 60 | NT-CC - NT-NO | -0.020754001 | 0.6743 | -0.09487 | 0.053361 |
| $\Delta W_{(3 day)}$ [cm] | 40 - 60 | NT-CC - ST-CC | -0.016111982 | 0.7439 | -0.09023 | 0.058003 |
| $\Delta W_{(3 day)}$ [cm] | 40 - 60 | NT-CC - ST-NO | -0.045683363 | 0.3618 | -0.1198 | 0.028431 |
| $\Delta W_{(3 day)}$ [cm] | 40 - 60 | NT-NO - ST-CC | 0.004642019 | 0.9249 | -0.06947 | 0.078757 |
| $\Delta W_{(3 day)}$ [cm] | 40 - 60 | NT-NO - ST-NO | -0.024929362 | 0.6143 | -0.09904 | 0.049185 |
| $\Delta W_{(3 day)}$ [cm] | 40 - 60 | ST-CC - ST-NO | -0.029571381 | 0.5509 | -0.10369 | 0.044543 |
| $\Delta W_{(3 day)}$ [cm] | 60 - 100 | NT-CC - NT-NO | -0.054132329 | 0.5927 | -0.20564 | 0.097372 |



| Variable [unit] | Depth Range [cm] | Comparison | Difference | P-value | LCL | UCL |
|---|---|---|---|---|---|---|
| $\Delta W_{(3\,day)}$ [cm] | 60 - 100 | NT-CC - ST-CC | -0.025543386 | 0.7998 | -0.17705 | 0.125961 |
| $\Delta W_{(3\,day)}$ [cm] | 60 - 100 | NT-CC - ST-NO | -0.088419413 | 0.3871 | -0.23992 | 0.063085 |
| $\Delta W_{(3\,day)}$ [cm] | 60 - 100 | NT-NO - ST-CC | 0.028588943 | 0.7766 | -0.12292 | 0.180094 |
| $\Delta W_{(3\,day)}$ [cm] | 60 - 100 | NT-NO - ST-NO | -0.034287083 | 0.7338 | -0.18579 | 0.117218 |
| $\Delta W_{(3\,day)}$ [cm] | 60 - 100 | ST-CC - ST-NO | -0.062876027 | 0.5353 | -0.21438 | 0.088629 |
| Pores: <0.2 µm | 0 - 5 | NT-CC - NT-NO | 0.002815389 | **0** | 0.002477 | 0.003153 |
| Pores: <0.2 µm | 0 - 5 | NT-CC - ST-CC | 0.002585408 | **0** | 0.002247 | 0.002924 |
| Pores: <0.2 µm | 0 - 5 | NT-CC - ST-NO | 0.002586301 | **0** | 0.002248 | 0.002924 |
| Pores: <0.2 µm | 0 - 5 | NT-NO - ST-CC | -2.30E-04 | 0.4768 | -5.68E-04 | 1.08E-04 |
| Pores: <0.2 µm | 0 - 5 | NT-NO - ST-NO | -2.29E-04 | 0.4803 | -5.67E-04 | 1.09E-04 |
| Pores: <0.2 µm | 0 - 5 | ST-CC - ST-NO | 8.93E-07 | 1 | -3.37E-04 | 3.39E-04 |
| Pores: 0.2-10 µm | 0 - 5 | NT-CC - NT-NO | -0.001562704 | **0.0144** | -0.00266 | -4.66E-04 |
| Pores: 0.2-10 µm | 0 - 5 | NT-CC - ST-CC | -6.15E-04 | 0.6364 | -0.00171 | 4.82E-04 |
| Pores: 0.2-10 µm | 0 - 5 | NT-CC - ST-NO | -0.00744646 | **0** | -0.00854 | -0.00635 |
| Pores: 0.2-10 µm | 0 - 5 | NT-NO - ST-CC | 9.47E-04 | 0.2625 | -1.49E-04 | 0.002044 |
| Pores: 0.2-10 µm | 0 - 5 | NT-NO - ST-NO | -0.005883755 | **0** | -0.00698 | -0.00479 |
| Pores: 0.2-10 µm | 0 - 5 | ST-CC - ST-NO | -0.006831157 | **0** | -0.00793 | -0.00573 |
| Pores: 10-50 µm | 0 - 5 | NT-CC - NT-NO | -0.002561805 | **0** | -0.00305 | -0.00207 |
| Pores: 10-50 µm | 0 - 5 | NT-CC - ST-CC | -0.00292541 | **0** | -0.00342 | -0.00243 |
| Pores: 10-50 µm | 0 - 5 | NT-CC - ST-NO | -0.003151266 | **0** | -0.00364 | -0.00266 |
| Pores: 10-50 µm | 0 - 5 | NT-NO - ST-CC | -3.64E-04 | 0.4018 | -8.56E-04 | 1.29E-04 |
| Pores: 10-50 µm | 0 - 5 | NT-NO - ST-NO | -5.89E-04 | **0.0572** | -0.00108 | -9.66E-05 |
| Pores: 10-50 µm | 0 - 5 | ST-CC - ST-NO | -2.26E-04 | 0.7663 | -7.19E-04 | 2.67E-04 |
| Pores: 50-1000 µm | 0 - 5 | NT-CC - NT-NO | 0.001801894 | **0** | 0.00127 | 0.002334 |
| Pores: 50-1000 µm | 0 - 5 | NT-CC - ST-CC | 0.001597245 | **0** | 0.001065 | 0.002129 |
| Pores: 50-1000 µm | 0 - 5 | NT-CC - ST-NO | 0.003943534 | **0** | 0.003411 | 0.004476 |
| Pores: 50-1000 µm | 0 - 5 | NT-NO - ST-CC | -2.05E-04 | 0.8483 | -7.37E-04 | 3.27E-04 |
| Pores: 50-1000 µm | 0 - 5 | NT-NO - ST-NO | 0.00214164 | **0** | 0.00161 | 0.002674 |
| Pores: 50-1000 µm | 0 - 5 | ST-CC - ST-NO | 0.002346288 | **0** | 0.001814 | 0.002878 |
| Pores: <0.2 µm | 20 - 25 | NT-CC - NT-NO | 3.40E-04 | **0** | 2.94E-04 | 3.87E-04 |
| Pores: <0.2 µm | 20 - 25 | NT-CC - ST-CC | 8.57E-04 | **0** | 8.10E-04 | 9.03E-04 |
| Pores: <0.2 µm | 20 - 25 | NT-CC - ST-NO | 8.68E-04 | **0** | 8.22E-04 | 9.15E-04 |
| Pores: <0.2 µm | 20 - 25 | NT-NO - ST-CC | 5.16E-04 | **0** | 4.70E-04 | 5.63E-04 |
| Pores: <0.2 µm | 20 - 25 | NT-NO - ST-NO | 5.28E-04 | **0** | 4.81E-04 | 5.74E-04 |
| Pores: <0.2 µm | 20 - 25 | ST-CC - ST-NO | 1.14E-05 | 0.9543 | -3.50E-05 | 5.79E-05 |
| Pores: 0.2-10 µm | 20 - 25 | NT-CC - NT-NO | 0.008308039 | **0** | 0.007172 | 0.009444 |
| Pores: 0.2-10 µm | 20 - 25 | NT-CC - ST-CC | 0.003161296 | **0** | 0.002025 | 0.004298 |
| Pores: 0.2-10 µm | 20 - 25 | NT-CC - ST-NO | 0.00445459 | **0** | 0.003318 | 0.005591 |
| Pores: 0.2-10 µm | 20 - 25 | NT-NO - ST-CC | -0.005146743 | **0** | -0.00628 | -0.00401 |
| Pores: 0.2-10 µm | 20 - 25 | NT-NO - ST-NO | -0.003853449 | **0** | -0.00499 | -0.00272 |
| Pores: 0.2-10 µm | 20 - 25 | ST-CC - ST-NO | 0.001293294 | **0.0772** | 1.57E-04 | 0.00243 |
| Pores: 10-50 µm | 20 - 25 | NT-CC - NT-NO | -1.08E-04 | 0.9915 | -8.98E-04 | 6.82E-04 |
| Pores: 10-50 µm | 20 - 25 | NT-CC - ST-CC | -8.86E-04 | **0.0841** | -0.00168 | -9.65E-05 |
| Pores: 10-50 µm | 20 - 25 | NT-CC - ST-NO | -2.75E-04 | 0.8822 | -0.00106 | 5.15E-04 |
| Pores: 10-50 µm | 20 - 25 | NT-NO - ST-CC | -7.78E-04 | 0.1601 | -0.00157 | 1.17E-05 |
| Pores: 10-50 µm | 20 - 25 | NT-NO - ST-NO | -1.67E-04 | 0.9702 | -9.57E-04 | 6.23E-04 |
| Pores: 10-50 µm | 20 - 25 | ST-CC - ST-NO | 6.12E-04 | 0.3581 | -1.78E-04 | 0.001401 |
| Pores: 50-1000 µm | 20 - 25 | NT-CC - NT-NO | -0.004292887 | **0** | -0.00494 | -0.00365 |



| Variable [unit] | Depth Range [cm] | Comparison | Difference | P-value | LCL | UCL |
|---|---|---|---|---|---|---|
| Pores: 50-1000 μm | 20 - 25 | NT-CC - ST-CC | -0.001873353 | **0** | -0.00252 | -0.00123 |
| Pores: 50-1000 μm | 20 - 25 | NT-CC - ST-NO | -0.001031633 | **0.0043** | -0.00168 | -3.87E-04 |
| Pores: 50-1000 μm | 20 - 25 | NT-NO - ST-CC | 0.002419534 | **0** | 0.001775 | 0.003064 |
| Pores: 50-1000 μm | 20 - 25 | NT-NO - ST-NO | 0.003261254 | **0** | 0.002617 | 0.003906 |
| Pores: 50-1000 μm | 20 - 25 | ST-CC - ST-NO | 8.42E-04 | **0.0304** | 1.97E-04 | 0.001486 |

**Data availability**

The soil data used for this study are available at [*will be deposited on Figshare.com upon acceptance of the manuscript for publication*].

**Author contribution**

All co-authors contributed to the study design. Sampling was carried out by SA and TA. Soil analysis and modeling were carried out by SA with supervision from TA. SA prepared the manuscript with contributions from all co-authors.

**Acknowledgments**

This work was made possible with support from the Conservation Agriculture Systems Project and the California Department of Water Resources.



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
