# Peer review of "Long-Term Impact of Cover Crop and Reduced Disturbance Tillage on Soil Pore Size Distribution and Soil Water Storage"

_SOIL, 2021_

## Author Response (AR1)

**Author's Responses**

Soil-2021-41
* * *
We thank the editor and both anonymous referees for their valuable comments on our manuscript. In this revised version we have included all comments and suggestions proposed by the Topical Editor as well as the referees.

The following is a point-by-point response to all comments. Please also note that we have made several other non-major changes and revisions to improve the manuscript in the spirit of these comments.

Sincerely,

Samuel Araya on behalf of all co-authors.
* * *
**Response to comments from the Topical Editor**

I agree with your suggested changes to the reviewer comments.

In addition, I think that you need to step down from statements about 'statistical significance'. According to the American Statistical Association, terms like 'statistical significant' and the use of a dichotomous interpretation of p-values should be avoided. Therefore, instead of referring to solely the 'statistical significance' of some effects including a p-value threshold, it is better to refer to the magnitude of the effect itself since this is what is important and add in parentheses the p-value of this effect. For instance the discussion on lines 330-346 starts with writing that the magnitude of the effects is small (but doesn't mention what the magnitude is) but then continues with an elaborated discussion about the statistical significance about the effects. In fact, this discussion is meaningless since the main point of interest is the effect. If the effect is too small to be of practical relevance, then it is not important anymore whether it is 'statistically significant' or not. Therefore, the discussion in this paragraph should focus on the magnitude of the effects and p-values of the effects should be mentioned so that the readers can interpret the effects and compare them with the variability and uncertainty that is associated to them. I brought up this example on lines 330-346 that clearly illustrates the issue but I propose that you consider this also in parts of the manuscript. I attached references to two papers about p-values and proposals to proscribe the use of terms like 'statistical significant' in scientific papers. These papers give you more information about the issue and the problem and I hope they help you revising the manuscript as well as addressing concerns raised by reviewer #1. It must be mentioned that these guidelines have been adopted already by several journals.

Stuart H. Hurlbert, Richard A. Levine & Jessica Utts (2019) Coup de Grâce for a Tough Old Bull: "Statistically Significant" Expires, The American Statistician, 73:sup1, 352-357, DOI: 10.1080/00031305.2018.1543616

Ronald L. Wasserstein & Nicole A. Lazar (2016) The ASA Statement on p-Values: Context, Process, and Purpose, The American Statistician, 70:2, 129-133, DOI: 10.1080/00031305.2016.1154108

> **We have thoroughly revised the manuscript to remove statements and focus on statistical significance and instead focused more on magnitude. We believe the revisions we made better communicate the findings accurately. Thank you for this very important constructive comment.**

**Response to Anonymous Referee #1**

**General comments**

This paper presents measurements of topsoil hydraulic properties made in four treatments of a long-term field experiment in California (with/without cover crops, conventional till/no-till) and uses these properties to simulate short-term soil water storage changes following subsurface drip irrigation.

The authors conclude that the no-till + cover crop system improves soil water storage (and presumably also the supply of water to the crop). However, this conclusion does not seem too well supported by the data as many of the differences between the treatments appear to be small. Many times in the paper, the authors report *p* values > 0.05 as being statistically significant, which is not standard practice. I am speculating now, but I can imagine the authors may have been worried that negative results and conclusions would not be considered publishable. In fact, I think that this kind of publication bias may be quite common. If so, it would be unfortunate because it would mean that our consensus view on the efficacy of different management practices to improve soil health could be somewhat biased.

> **We chose to use a less conservative p-value (p < 0.15) as our standard in the significance tests to make note of differences that were more abundant in our research at that level. However, the point is well taken that such p > 0.1 is not standard practice in the field. We have decided to address this concern by first placing a paragraph in the methods that explicitly communicates and highlights this. Furthermore, throughout the manuscript, we will briefly make this note where we reference these higher p-values.**

The small number of replicates (4) is an important limitation of this study. Forward modelling based on such limited replication, without the benefit of field monitoring data to calibrate and validate the model is very uncertain. These uncertainties are not discussed by the authors. Field measurements of soil water contents/potentials during and after irrigation would have helped to strengthen the study.

> **We decided on 4 replicates for two reasons, (1) the time-consuming nature of the hydraulic property measurements, and (2) the relatively small variance observed among the replicates in relation to the treatments (for example, visual inspection of the retention and conductivity curves). However, to address the concerns of small replicate count, we will highlight the replicate count in the results and add a paragraph in the discussion section where we discuss the study's limitations.**

More details on the soil properties and the tillage practices implemented at the site are also needed (points 6-8 under "Specific comments"). The routines in the model related to root water uptake and soil evaporation also need to be better described (see points 10 and 11 under "Specific comments")

> **Thank you, we have responded to the comments in the specific comments section.**

The paper is generally well written and easy to read. There are a number of grammatical errors in addition to the ones noted below under "Technical corrections", but these will be easy to correct.

> **Thank you for taking the time to improve the paper. We will carefully address the technical comments you provided.**

**Specific comments**

**1-1.** Line 18, and lines 20-22: these statements appear to be contradictory. First, the authors write that the differences in water storage are marginal, but then they conclude that NT and CC systems improve water retention at the field scale. This should be clarified.

> **We have clarified the statement by explicitly stating that the practices show beneficial effects in terms of PSD changes and show marginal improvements in water storage. The abstract ending now reads,** *"The study concludes that the long-term practices of NT and CC systems were beneficial in terms of changes to the PSD. NT and CC systems also made marginal improvements in soil water conductivity and water storage, improving water retention at the plot scale."*

**1-2.** Line 44: The authors should write this in a less categorical way, by replacing "have been shown" to "may", because it is definitely not always the case that reduced or no-till systems increase carbon sequestration in soil (see Meurer et al 2018). This is probably mostly because under some agro-environmental conditions, crop growth is poorer under no-till (Pittelkow et al., 2015), which reduces the carbon inputs to soil.

> **We accept this comment and have re-write the entire paragraph to reflect the suggested points. The quoted sentence now start as, "***Several studies have shown, for example, that reduced disturbance tillage systems .. ***".**

**1-3.** Lines 47-48: Yield certainly is sometimes compromised by no-till (see point 2 above, Pittlekow et al., 2015). Please re-phrase this to acknowledge this fact.

> *We have replaced "*Without compromising yield and reducing cost.*" **with** *"while some studies show that reduced disturbance tillage reduce yield (Pittlekow et al., 2015) others have found that the yield is unaffected  (Naab et al., 2017; Rasmussen, 1999; Alvarez and Steinbach, 2009)"*

**1-4.** Line 60: 900 mm annual rainfall?

> **Change accepted.**

**1-5.** Lines 61-62: Perhaps you should explain why no-till can exacerbate compaction problems (because the soil is not loosened, but it is still trafficked) … and also that this can impact yields negatively (Pittelkow et al., 2015)?

> **We have update the second sentence as follows: "***Without tillage to loosen the soil, reduced tillage systems can cause soil consolidation…***". We also have added the reference.**

**1-6.** Lines 78-79: it's only a more descriptive term if no-till really was adopted at the site. This is not fully clear to me. At line 75, you write "reduced disturbance". It's important to be clear and explicit about the tillage system. Is it "reduced tillage" or "no-till"? The tillage operations should be described in more detail for both ST and NT, giving information on the time of year, the implements used, and the depths to which they are operating.

> **We believe that the term "no-tillage," which relies on procedures that enable crop planting directly into the soil with no primary or secondary tillage since harvest of the previous crop (SSSA, 1996), most aptly characterizes this tillage system and is a better descriptor than alternatives such as "reduced," "minimum," or "conservation tillage" that have been used previously  (Reicosky, 2015; Mitchell et al., 2019). We are aware of the confusing and vague language that is sometimes associated with tillage practices and the need for precision when**

**reporting tillage systems in scientific work. Therefore we try to be clear with the language we use.**

**At line 75, we have replaced** *"reduced disturbance tillage"* **with** *"no-till"***.**

**Regarding the detailed description of the systems, we decided to summarize the management practices and not include detailed management practices as this has been described in multiple previous publications. However, we agree that a more detailed summary is necessary. Following line 83 we have added the following summary.** *"Both the ST and the NT systems were previously described in detail (example, Veenstra et al., 2006; Mitchell et al., 2015). The NT systems were managed from the principle of reducing primary intercrop tillage to the greatest extent possible. Controlled traffic farming practices that restrict tractor traffic to certain furrows were used, and planting beds were not moved or destroyed in these systems during the entire study period. The only soil disturbance operations used in the NT systems were shallow cultivation during the first eight years of the project, since 2012. However, the only soil disturbance occurs at the time of seeding or transplanting. The ST systems consisted of multiple conventional intercrop tillage operations which break down and establish new beds following harvest and represent the normal operations of the San Juaquin Valley in terms of intensity, depth, and timing of tillage."*

> **Reicosky, E.C. 2015. Conservation tillage is not conservation agriculture. J. Soil water Conserv 70(5):103-107.**
>
> **Mitchell, J.P., D.C. Reicosky, E.A. Kueneman, J. Fisher, and D. Beck. 2019. Conservation agriculture systems. CAB Reviews 14(001):1-25. https://www.cabi.org/cabreviews/ review/20193184383**
>
> **Mitchell, J.P., A. Shrestha, D.S. Munk, and K.J. Hembree. 2015. Cotton response to long-term no-tillage and cover cropping in the San Joaquin Valley. J. Cotton Science. 19:1-10.**
>
> **Soil Science Society of America. 1996. Glossary of Soil Science Terms. Soil Science Society of America, Madison, WI.**
>
> **Veenstra, Jessica J., William R. Horwath, and J.P. Mitchell. 2007. Tillage and cover cropping effects on aggregate-protected carbon in cotton and tomato. Soil Sci. Soc. Am. J. 71:362-371.**

**1-7.** Lines 81-83: it's important to mention that the tractor traffic was controlled in this way, but what about harvesters? Presumably this kind of traffic was not controlled in the same way?

> **All traffic was controlled in this manner, including harvesters. Please see the response to comment 1-6 above.**

**1-8.** Lines 84-87: The authors must give more information on the basic soil properties at the site: at the minimum, information is needed on the particle size distribution and organic carbon contents (the latter specified for each treatment, as they presumably differ after 20+ years).

> **We have included descriptions of particle size distribution and organic carbon contents. We also added a reference to previous studies that have a more detailed description of our site's soils.**

**1-9.** Lines 128-129: Why did you calculate unsaturated K at -10 kPa, and not say, at -33 kPa (field capacity)? You should also say how you did this: by fitting to the HYPROP data presumably?

**We have changed "*calculate*" to "*compared*". We also describe that we read these values from the fitted hydraulic data.**

**We compared K at -10 kPa because in field conditions, soils rarely achieve 100 % saturation because of air entrapment and other factors. We assumed -10 kPa would be a better scale to represent field infiltration conditions.**

**1-10.** Line 174: What does "… a radius of maximum uptake intensity at 0 cm" mean? Not all readers will be familiar with this 3D water uptake model, so this should be explained better.

**We have re-writen to clarify the variables as:** *"… using Vrugt et al. (2001) function. Values for the required variables are given in table 1"*

| Variable | Value (cm) |
|---|---|
| *Maximum rooting depth* | *35* |
| *Maximum rooting radius* | *15* |
| *Depth of maximum uptake intensity* | *10* |
| *Radius of maximum uptake intensity* | *0 (at center)* |

**1-11.** Lines 175-182: this entire section is unclear to me. The authors mention evapotranspiration, but it's not clear how soil evaporation and transpiration are dealt with individually. For example, the name Feddes is mentioned on line 172, so I presume that the Feddes model is used to calculate actual transpiration from the potential uptake rate, but this should be explicitly stated. But how is soil evaporation reduced below the potential rate (which I presume is given by equation 4) when the soil surface dries out?

**The HYDRUS model only requires input of potential evapotranspiration. The model calculates actual evapotranspiration during simulation depending on the instantaneous soil moisture and root water uptake conditions.**

**We have clarified this section as follows:** *"The atmospheric boundary condition was defined by potential crop evapotranspiration (ETc) which was calculated from potential evapotranspiration (ETo) and a crop coefficient (Kc) (Equation 4).*

*Hourly ETo for a week in May 2018 (6-12 of May) was retrieved from the nearest weather station…."*

**1-12.** Line 227 and elsewhere in the paper: the authors refer to results with *p* values greater than 0.05 as "statistically significant" (or similar phrases). The authors should reserve such a description for results with $p < 0.05$. Instead, write "… a tendency for …" or something similar.

**We have now used similar phrases as suggested by the reviewer and indicate the actual p values wherever appropriate. However, we still believe that using p<0.15 as a cutoff for comparisons in the figures is useful and communicates the results clearly.**

**1-13.** Line 237: "… healthy organic matter cycling" is vague. This should be written more specifically.

**We have removed the word 'healthy', which we agree may not be as precise. We have re-write as follows:** *"…soil processes including soil organic matter cycling …"*

**1-14.** Lines 246: Jarvis (2007) did not discuss these processes. It would be better to cite the recent review on soil structure dynamics by Meurer et al (2020) here instead.

> **Thank you for suggesting a good reference. We have added Meurer et al. (2020) to the list of citations for this sentence. However, we believe Jarvis (2007) is also relevant that review paper also has relevant discussions and sources on the importance of plant roots in relation to soil structure.**

**1-15.** Figure 7. Is the y-axis on the Kunsat plot correct? 10 to the power of 0.004 is only 1.009. Isn't the variation in K larger than 1 to 1.009 cm/d?

> **Upon further consideration based on this comment, we have decided that Figure 7b is not necessary and have removed it from the manuscript. In the text, references to this figure, lines 274-277 have ben appropriately removed**

**Technical corrections**

**1-16.** Lines 45, 47 and 52: terms like "soil fertility" and "environmental quality" are rather vague. Please replace these with terms specific to what was measured in these studies you cite.

> **We have gone through the entire paragraph and replace those terms with the actual soil and environmental property mentioned in the references.**

**1-17.** Line 122: add "of water tensions" after "range"

> **Comment accepted.**

**1-18.** Line 124: Write … "We define field capacity …" (this definition is only conventional in the U.S., not worldwide)

> **Comment accepted.**

**1-19.** Line 125: delete the minus signs prior to 33 and 1500 (you refer to suction)

> **Comment accepted.**

**1-20.** Lines 130-131: this can be deleted, as it is defined in connection to equation 1.

> **We agree that it is defined in connection to the equation, however, we believe restating the terms again in this way is helpful and that this sentence remains in the manuscript.**

**1-21.** Line 163: delete "that of"

> **Comment accepted.**

**1-22.** Line 165: "the van Genuchten-Mualem hydraulic model (van Genuchten, 1980)"

> **Comment accepted.**

**1-23.** Line 171: add… "($S_r$ in in equation 3)" after "root water uptake"

> **Comment accepted.**

**1-24.** Line 175: insert "An .. " before "… atmospheric .."

> **Comment accepted.**

**1-25.** Lines 208-210: this sentence can be deleted (it's repeating the methods). The authors should refer to figure 4 here, but in a different way … "Figure 4 shows one example of …"

**We accept this comment. The paragraph now reads as,** *"An example of water conductivity and retention measurement for a single soil sample is shown in Figure 4."*

**1-26.** Line 227: Replace "A unique …" by "One …"

**Comment accepted.**

**1-27.** Lines 238-239: this is unclear. Is there text missing here?

**We have re-writen the statement and correct the type to read,** *"ST-NO plots had the lowest relative abundance of larger macropores (50 – 1000 μm) while NT-CC had the highest proportion (Figure 5B)"*

**1-28.** Line 260: section 0?

**Typo fixed as,** *"section 3.1"*

**1-29.** Line 327: insert "of drainage" after "days"

**Comment accepted.**

**1-30.** Line 364: replace "steady-state" by "equilibrium"

**Comment accepted.**

**Responses to Anonymous Referee #2**

The manuscripts explore the effect of long-term reduced tillage and cover crops on soil hydraulic properties. Based on laboratory samples from a long-term experiment, they derived the water retention curve (WRC) parameters and the K(h). Using these observations, they simulate in Hydrus 2D, the effect of an irrigation even and analyze the dynamic of the water storage that follows.

The fact that the samples came from a long-term experiment with a sound statistical design is a strength of the research. The analysis of the observed hydraulic parameters is also statistically sound. The use of a simulation to access some more dynamic information on the soil profiles is interesting. While I find it less valuable than direct field observations, it remains based on laboratory measurements (K(h) and WRC). As such, it helps to assess the dynamics of water storage in the soil which is a valuable insight for the interpretation even if not confirmed by field measurements. The authors also discuss their results at the light of other works and seem well-informed of the work done in the area.

In general the manuscript is well written, the figures are clear and well-designed with good caption. The errors bars are used to display uncertainty whenever possible. The abstract would gain to be a bit reworked so that the message and the findings of the work came through better. Also in several places, a bit more discussion is needed or further interpretation and hypothesis will deepen the discussion. Also, a discussion about the limitation of the modeling approach is needed.

> **Thank you for the comments. We have made several improvements throughout the manuscript based on your specific comments. We have improved the abstract for readability and to better communicate the findings. We have also add a paragraph in the discussion section that highlights the study's limitations more clearly, including the limitations of the modeling approach and other limitations pointed out by Referee #1. Please find the many changes we made in response to your specific comments below.**

**General comments**

The advantage of the simulation and its links with the observation doesn't come through easily as we read. It's difficult to see straight from the start why the simulations are needed and what they bring to the story. Maybe it can help to highlight, at the end of the introduction, the limitation of traditional PAW and field capacity to better highlight the need for simulation?

As mentioned by the authors, tillage can temporarily impact soil hydraulic parameters and soil properties. Potentially doesn't the growth of the (cover) crops also have an impact on WRC and K(h)? Given the study provides topsoil and subsoil samples, can you discuss more the expected dynamics of soil hydraulic properties expected from each depth through the growing season? A more global discussion on the effect of the sampling depth would be great.

> **Thank you for the suggestion. We have expand our discussion on this in our response to your specific comment (2-19) for example, we have expanded such discussion and added multiple references.**

The simulations bring insights into the dynamic water storage of the soil columns. However, I think it should be pointed out that these simulations are not backed up by observations of water content in the soil profile but only based on laboratory derived WRC and K(h).

**Specific comments**

   **2-1.** L14: "an improved structure" -> put it more explicitly: larger proportion of smaller pores?

> **We believe stating it in a more generalized way** (*"improved soil structure in terms of pore size distribution"*) **better summarizes our findings than the more detailed suggestion. Furthermore, the improvement of structure is due to larger proportion of smaller pores and the bi-modal nature of the distribution.**

**2-2.** L15: re-phrase more simply: "The conventional measurement of water content at field capacity (water content at -33 kPa suction) and the associated plant available water (PAW) showed that NT and CC plots had lower water content at field capacity and lower PAW compared to standard-till (ST) and plots without cover crop (NO)" -> "NT and CC plots had lower water content at field capacity (-33 kPa suction) and lower plant available water (PAW) compared to standard-till (ST) and plots without cover crop (NO)"

> **We have accepted the recommendation as is.**

**2-3.** L18: "higher profile-level water storage"? -> what does 'profile-level' mean? maybe re-phrase more simply? (after reading the paper, I realized it was referring to the simulated provides but I didn't get it at first read, maybe drop 'profile-level'? You've already said it was in simulations)

> **We deleted the word 'profile-level' as suggested.**

**2-4.** fig1: missing north arrow

> **We added a north arrow to Figure 1.**

**2-5.** L60: < 9000 mm annual rainfall is drier? -> 900 in original paper

> **The typo is now corrected as suggested.**

**2-6.** L65: 'individual and interactive impact'? -> 'individual impact and its interaction

> **Comment accepted.**

**2-7.** L74: please also give GPS coordinates

> **We have now included the geographic coordinates in parenthesis.**

**2-8.** L79: I though no-till was more absolute than reduced tillage. In reduced tillage, as you explain, there is a reduction of tillage operation, but no-till really means no-till no? If its a reduced tillage, maybe RT would be a better acronym. Also, what is the tillage depth? Does the reduced tillage involve a non-inversion tillage? What tools was used for the tillage (rotovator, harrow, chisel plow, moldboard plow) more explanation is needed here even if more details are published in Mitchell et al. previous papers.

> **We believe that the term "no-tillage," which relies on procedures that enable crop planting directly into the soil with no primary or secondary tillage since harvest of the previous crop (SSSA, 1996), most aptly characterizes this tillage system and is a better descriptor than alternatives such as "reduced," "minimum," or "conservation tillage" that have been used previously (Reicosky, 2015; Mitchell et al., 2019). We are aware of the confusing and vague language that is sometimes associated with tillage practices and the need for precision when reporting tillage systems in scientific work. Therefore we try to be clear with the language we use.**
>
> **Regarding the detailed description of the systems, we decided to summarize the management practices and not include detailed management practices as this has been described in multiple previous publications. However, we agree that a more detailed summary is necessary. Following line 83 we have add the following summary that addresses the points**

**raised in your comment.** *"Both the ST and the NT systems were previously described in detail (example, Veenstra et al., 2006; Mitchell et al., 2015). The NT systems were managed from the principle of reducing primary intercrop tillage to the greatest extent possible. Controlled traffic farming practices that restrict tractor traffic to certain furrows were used and planting beds were not moved or destroyed in these systems during the entire study period. The only soil disturbance operations used in the NT systems were shallow cultivation during the first eight years of the project, since 2012, however, the only soil disturbance occurs at the time of seeding or transplanting. The ST systems consisted of multiple conventional intercrop tillage operations which break down and establish new beds following harvest and represent the normal operations of the San Juaquin Valley in terms of intensity, depth, and timing of tillage."*

> **Reicosky, E.C. 2015. Conservation tillage is not conservation agriculture. J. Soil water Conserv 70(5):103-107.**
>
> **Mitchell, J.P., D.C. Reicosky, E.A. Kueneman, J. Fisher, and D. Beck. 2019. Conservation agriculture systems. CAB Reviews 14(001):1-25. https://www.cabi.org/cabreviews/ review/20193184383**
>
> **Mitchell, J.P., A. Shrestha, D.S. Munk, and K.J. Hembree. 2015. Cotton response to long-term no-tillage and cover cropping in the San Joaquin Valley. J. Cotton Science. 19:1-10.**
>
> **Soil Science Society of America. 1996. Glossary of Soil Science Terms. Soil Science Society of America, Madison, WI.**
>
> **Veenstra, Jessica J., William R. Horwath, and J.P. Mitchell. 2007. Tillage and cover cropping effects on aggregate-protected carbon in cotton and tomato. Soil Sci. Soc. Am. J. 71:362-371.**

**2-9.** L97: "months after tillage" -> how many months about?

> **We have add the number of months (i.e., over 5-months).**

**2-10.** fig2a: different colors of layers but no legend, also why a 3D schematic if the modeling is only in 2D, this is confusing, I would do a 2D schematic if only 2D modeling

> **We used a gradient of colors to highlight the soil layers used in the model. The colors serve only a cosmetic function.**
>
> **We show a 3D schematic because the simulation domain was 3D axisymmetric cylinder. HYDRUS-2D is capable of performing simulations with this type of 3D domain.**

**2-11.** fig2b: why no flux on the side? If it's in the field, it should be free flow, no? Unless you simulate your cylindrical lysimeter

> **The sides were set to no-flux as the radius of 18 cm is half the distance of the emitters spacing. Therefore, the midway between the drip lines is a flow divide, where flow lines from adjacent drippers meet. In other words, this boundary represents a symmetric condition that isolates the smallest representative hydrologic unit in the plot.**

**2-12.** L188: why the spin-off has only 2.5 irrigation compared to the real simulation?

> **The spin-off period was instituted to make sure that the starting conditions of the different soils are closer to what they would be in nature. The selection of 2.5 cm irrigation was to create a condition that represents the differences of the soil structure but also results in a drier starting condition. This combination allows exploring a wider range of water dynamics**

**without simulating multiple seasons. Note that the goal of these simulations was to compare the effect of the treatments on water conservation and water use efficiency, not necessarily to represent a particular field scenario.**

**2-13.** L196: equivalent water depth (change?) -> not sure I understand this, is it the water in the entire profile or at a specific depth? what does the 'water depth' means?

**Equivalent water depth is the volume of irrigation water (cm$^3$), divided by the area (cm$^2$) to which this irrigation water is applied. This is the amount of water held in a profile in cm depth of water. We used water depth because it is a more conventional unit of measuring irrigation on farms and is a more convenient metric to compare with precipitation and ET.**

**2-14.** L205: seems sound. Do you also provide the .R file and the data as open-source attachment to the article?

**Yes. All the data and R code will be provided in a public repository.**

**2-15.** L238: 'all the treatments had higher relative... compared to ... had the highest proportion' -> re-phrase

**We have re-writen the sentence for clarity as follows,** *"ST-NO plots had the lowest relative abundance of larger macropores (50 – 1000 μm) while NT-CC had the highest proportion (Figure 5B)."*

**2-16.** L260: 'discussed at section 0' -> 'section 3.1?

**Comment accepted.**

**2-17.** fig7: why is NT-CC for Ks so much higher while there not such difference for K100cm? Discuss

**We have decided that Figure 7b is not necessary and have removed it from the manuscript. In the text, references to this figure, lines 274-277, are appropriately changed.**

**2-18.** L300: "Our results showed that while this was the case with ST, it was not the case for NT" -> do you have an explanation/hypothesis?

**We have added the following possible explanation after that statement,** *"This is possibly due to the relatively intact roots left after CC is removed, unlike the ST plots where the effect of CC roots is disturbed by tillage following the CC removal."*

**2-19.** L301: 'For the subsurface layer of NT treatments, θ FC was significantly lower for the NT-CC compared with NT-NO treatments.' -> could this be that the roots of the CC are reaching deeper down? what is your hypothesis? discuss more

**We agree with this comment. We have added the following discussion to clarify the point:**
*"For the subsurface layer of NT treatments, $\theta_{FC}$ was significantly lower for the NT-CC compared with NT-NO treatments. This difference suggests that roots from cover crops extend below our surface layer and have the potential to significantly alter soil structure. This subsurface effect of CC may be masked by frequent disturbance in the ST treatments. This observation is consistent with recent studies that have shown that the effect of cover crops extends below the so-called plough layer (rooting depth of approximately 30 cm) (Rath et al., 2021; Veloso et al., 2018; Sastre et al., 2018; Tautges et al., 2019)."*

**2-20.** fig10: are these curves for one single plot/simulation? or averaged over all 4 plots for each treatment? Make this clear in the caption.

**We have included a statement in the caption to explain that the lines are group averages.**

**2-21.** L344: "(P < 0.15)" -> lower p for p-value (typo)

**Comment accepted.**

**2-22.** fig11: could it be plotted the same way as figure 8 with both top and subsoil in the same subplot? that will make comparison easier

**We have updated figure 11 to be similar to figure 8 in layout.**

**2-23.** figA4 and A5 shows the distribution for a set of 4 plots but you simulate 4*4 in total right? are these plots average per treatment or just one pick among the 4 replicates? Make this clear in caption

**We have updated the caption to indicate that the values are treatment averages.**

---

## Editor Decision (ED1)

**Long-Term Impact of Cover Crop and Reduced Disturbance Tillage on Soil Pore Size Distribution and Soil Water Storage**

Samuel N. Araya1, Jeffrey P. Mitchell2, Jan W. Hopmans3, and Teamrat A. Ghezzehei4

1Earth System Science, Stanford University, Stanford, CA, USA

2Department of Plant Sciences, University of California, Davis, CA, USA
 3Department of Land, Air and Water Resources, University of California, Davis, CA, USA
 4Life and Environmental Science, University of California, Merced, CA, USA

Correspondence to: Samuel N. Araya (araya@stanford.edu)

[revised manuscript text omitted]

The effective pore size distribution (PSD) was estimated from the slope of the WRC using the differential water capacity (Klute, 1986). For this, the WRC— $\theta(h)$ —was first transformed into a curve of effective saturation (*S*) as a function of effective pore radius (*r*), *S*(*r*). *S* was calculated as  $S = (\theta - \theta_r)/(\theta_s - \theta_r)$ , where  $\theta_s$  and  $\theta_r$  are the saturated and residual volumetric water contents estimated from a bimodal constrained van Genuchten model fit (Durner, 1994) of measured WRC. The draining pore radius was approximated using the Young-Laplace equation (1):

$$r = \frac{2\gamma \cos\left(\beta\right)}{\rho_{w}gh} = \frac{1490}{h} \tag{1}$$

[revised manuscript text omitted]

| •        | Formatted: Caption, Keep with next                                                                                                                                                   |
|----------|--------------------------------------------------------------------------------------------------------------------------------------------------------------------------------------|
|          |                                                                                                                                                                                      |
| •        | Formatted: Normal                                                                                                                                                                    |
|          | Formatted Table                                                                                                                                                                      |
|          | Formatted: Normal                                                                                                                                                                    |
| •        | Formatted: Normal                                                                                                                                                                    |
|          |                                                                                                                                                                                      |
| •        | Formatted: Normal                                                                                                                                                                    |
|          |                                                                                                                                                                                      |
| •        | Formatted: Normal                                                                                                                                                                    |
|          |                                                                                                                                                                                      |
|          | Deleted: with a maximum rooting depth of 35 cm, a maximum rooting radius of 15 cm, depth of maximum uptake intensity at 10 cm and radius of maximum uptake intensity at 0 cm. |
| _ | eni, and radius of maximum uptake mensity at 0 eni.                                                                                                                                  |
|          |                                                                                                                                                                                      |

$$ET_c = K_c \times ET_0 \tag{4}$$

Where  $ET_c$  [LT-1]is potential crop evapotranspiration,  $K_c$  [-] is crop coefficient (= 1.15 for tomato mid-season (Allen et al., 1998), and  $ET_0$  [LT-1] is the reference potential evapotranspiration.

Hourly, reference potential evapotranspiration ( $ET_0$ ) for a week (May 6 to 12, 2018) were retrived from the nearest weather station (CIMIS Five Points Station, https://cimis.water.ca.gov/).

270

275

Figure 2 (A) 3D schematic representation of the domain geometry and material distribution. (B) Domain setup in Hydrus-2D showing the finite element mesh, related boundary conditions, and potential root water uptake rate distribution.

The starting pressure head of the entire model domain was set to -1000 cm, and simulation was initialized by a 14-week spinweeks after which the final simulation is run with 4.8 cm equivalent depth irrigation (at the rate of 0.6 cm h-1 for 8 h) application (Figure 3). The amount of water retained in a given soil profile layer following irrigation is calculated as equivalent water depth changes using Equation 5.

$$\Delta W_t = W_t - W_{t0} \tag{5}$$

where  $\Delta W_t$  [L] is equivalent water depth retained in the soil profile t hours after irrigation application,  $W_t$  is the equivalent 285 water depth in the soil profile t hours after irrigation, and  $Wt_0$  is equivalent water depth immediately before irrigation application.

---

## Author Response (AR2)

**Author's Responses to Comments from the Topical Editor**

Soil-2021-41

Thank you for the thoughtful comments and suggestions. We have accepted the majority of the recommendations you made. The following is a point-by-point response to all comments. All changes made in this iteration have been tracked in the Microsoft Word document.

Sincerely,

Samuel Araya on behalf of all co-authors.
* * *
**General Comments**

I think you responded well to all reviewer comments. I gave the paper another careful perusal and you find the comments in the attached pdf. Please check the units carefully and I propose to use pressure head consistently throughout the paper and avoid the medley of suction, tension, matric potential in kPa or cm, minus or positive. Also give some information about the statistical analyses of the simulation results and how the variability of the hydraulic functions was considered in the simulations.

> **We have thoroughly revised the manuscript again to address the issues mentioned. We have now corrected references to soil pressure head to be uniform throughout the manuscript. We have also clarified that the simulation was done for each replicate and results were analyzed on treatment average basis.**

**Point-by-point comments**

1. Line 15: check consistency with units used later in the text.

   > **We have made suction units uniform throughout the manuscript as suggested. In the abstract we prefer it stay as kPa because it is a standard unit of pressure and without explanation present in the body, kPa seems a better choice to us.**

2. Line 19: maybe better: hydraulic states? Hydraulic properties are the retention and conductivity curves?

   > **We have accepted the recommended change.**

3. Line 20: I could not follow the rationale behind this sentence. What is the difference between water retention and storage? Why does an improvement in storage improves retention? I would rather think it should be opposite: an improvement of retention improves the storage.

   > **We have clarified the statement by removing the reference to retention and just use storage. We were trying to explain the steady-state retention and the simulated dynamic storage.**

4. Line 134: This seems to be a repetition of the sentence highlighted above.

   > **We have removed the repeated sentence.**

5. Line 137: Above you write that the project started in 1999? Or should it be: Since 2012, however, the only soil disturbance occurs .... ?

**We have corrected the sentence. For clarity. It now reads:** *"From 2012 onwards, soil disturbing activity in the NT systems occurs only at times of seeding or transplanting."*

6. Line 143: I know you are living in the Golden State but these are densities of gold. I suppose the units are g kg-1?

   **We have corrected the units as suggested to realistic carbon numbers.**

7. Line 148: ha-1

   **Correction accepted.**

8. Lines 170, 182,185: I think it would be better to use pressure heads consistently through the paper. Now you are using tensions, suction, matric potential, heads throughout.

   **Throughout the manuscript, we have replaced all pressure measurements to suction in cm.**

9. Line 186: do you mean: recover from turgidity loss? Why should a plant recover from turgidity? When it wilts, it loses turgor.

   **We have corrected the statement to read:** *"limit beyond which plants cannot recover their turgidity."*

10. Line 189: This is rather 'near-field capacity'.

    **Comment accepted.**

11. Line 192: Move here.

    **We have moved the sentence as suggested.**

12. Line 216: the problem with including this is that the units of h (cm) are not consistent with the units of gamma, rho_w, and g. I would propose skipping it and write that the  factor 1490 is a factor that contains unit conversions, the surface tension of water, the mass density of water and the acceleration due to gravity. The contact angle was assumed to be 0.

    **Comment accepted, we have removed the middle equation and variable description.**

13. Line 219: I do not understand why you should have a minus sign here. I think this can be skipped.

    **We needed to multiply by -1 as the density function appears inverted because the saturation function is inversely related to pore size.**

14. Lines 220: Was the S(r) function fitted with a spline or the S(ln(r)) function. The latter would be more consistent, I think.

    **We have corrected it as suggested.**

15. Line 225 I could not find back whether you did more than one simulation for each treatment using hydraulic properties that were derived from different soil samples. Looking at table B1, I think you must have since you are also doing statistical tests on the simulated water contents and storages. It is important to include the information in the text that you did several simulations for each treatment and to write how the hydraulic functions for each simulation were derived.

**we have added a sentence to explain this information as:** *"We performed a total of 16 independent simulations, one for each treatment plot, using soil hydraulic properties that were derived from the respective treatment soil samples."*

16. Line 231: the gravity term is missing in the equation.

    **The effect of gravity is already captured in the pressure head change along the vertical coordinate space. The equation is taken from the HYDRUS-2D manual and represents how flow is simulated.**

17. Line 233: you have used h already for suction.

    **We have changed it to capital H to avoid inconsistency with the rest of the manuscript.**

18. Line 374: I suppose you mean section 3.1? "appear"

    **We have corrected the number and the typo.**

19. Line 413: This seems very high to me. It suggests that the majority of the rainfall would run off in the plots without CC, which seems quite unexpected. Or is the infiltration capacity increased by a factor 2.8?

    **The infiltration time was faster by a factor of 2.8. We have clarified the statement as:** *"an increase by a factor of 2.8 times compared to ..."*.

20. Lines 419, 477: 'apparent'. 'dimensional'

    **We have corrected the typos.**

21. Line 514: a

    **We have corrected the statement as:** *"there were no apparent differences..."*.

22. Line 563: independently. negligible.

    **Both typos have been corrected.**

23. Line 594: decrease for which treatment?

    **We have corrected the sentence as:** *"Suggested soils under NT-CC practices had less ability to store water."*

24. Line 603: see comment in the abstract.

    **We have corrected this statement in the same manner to the correction we made in the abstract.**

25. Line 605: delete "to".

    **Comment accepted.**

---

## Author Response (AR3)

**Authors' Responses to Comments from the Executive Editor and Topical Editor**

Soil-2021-41

Dear Dr. Quinton and Dr. Vanderborght,

Thank you for your consideration of our manuscript and comments. We have addressed the issue you raised on Eq. 3 as suggested. In the current version, we have revised the description for H as, *"H [L] is the total hydraulic head (sum of pressure head and elevation)"*.

Sincerely,

Samuel Araya on behalf of all co-authors.